# WHY IN-CONTEXT LEARNING MODELS ARE GOOD FEW-SHOT LEARNERS?

**Shiguang Wu**
Department of Electronic Engineering,
Tsinghua University
wsg23@mails.tsinghua.edu.cn

**Yaqing Wang**
Beijing Institute of Mathematical
Sciences and Applications
wangyaqing@bimsa.cn

**Quanming Yao**[*]
Department of Electronic Engineering, Tsinghua University
State Key laboratory of Space Network and Communications, Tsinghua University
qyaoaa@tsinghua.edu.cn

## ABSTRACT

We explore in-context learning (ICL) models from a learning-to-learn perspective. Unlike studies that identify specific learning algorithms in ICL models, we compare ICL models with typical meta-learners to understand their superior performance. We theoretically prove the expressiveness of ICL models as learning algorithms and examine their learnability and generalizability. Our findings show that ICL with transformers can effectively construct data-dependent learning algorithms instead of directly follow existing ones (including gradient-based, metric-based, and amortization-based meta-learners). The construction of such learning algorithm is determined by the pre-training process, as a function fitting the training distribution, which raises generalizability as an important issue. With above understanding, we propose strategies to transfer techniques for classical deep networks to meta-level to further improve ICL. As examples, we implement meta-level meta-learning for domain adaptability with limited data and meta-level curriculum learning for accelerated convergence during pre-training, demonstrating their empirical effectiveness.

## 1 INTRODUCTION

Large Language Models (LLMs) (Achiam et al., 2023) have witnessed remarkable progress in recent years. Beyond traditional natural language processing tasks such as machine translation and sentiment analysis, LLMs have gained prominence in solving more complex tasks by understanding instructions and examples from human input and generating coherent, human-like text. LLMs use in-context learning (ICL) (Brown, 2020) to understand and generate responses based on the input text. Given a prompt containing examples (input-output pairs) from a task and a query input, ICL allows the LLM to generate the corresponding output without altering their weights. For example, given "*happy -> positive; sad -> negative; blue ->*", the model can output "*negative*", while given "*green -> cool; yellow -> warm; blue ->*" the model can output "*cool*". Formally, ICL can be formulated as follows: given input $(\mathbf{x}^{(1)}, \mathbf{y}^{(1)}, \cdots, \mathbf{x}^{(n)}, \mathbf{y}^{(n)}, \mathbf{x}^{(n+1)})$, where there is an underlying task $f$ such that $\mathbf{y}^{(i)} = f(\mathbf{x}^{(i)})$, the model outputs a prediction of $f(\mathbf{x}^{(n+1)})$. By pre-training to simulate the above behavior over a distribution of $f$, the ICL model can generalize to unseen tasks.

The remarkable performance of LLMs across a wide range of applications has garnered significant attention, to understand how the ICL ability is acquired and executed. However, ICL has thus far been well understood primarily in highly simplified settings: linear-transformers trained on linear regression tasks. In these cases, the model is shown to precisely learn to perform pre-conditioned gradient descent based on input examples, with explicit weights corresponding to the global minimum during pre-training (Von Oswald et al., 2023; Mahankali et al., 2024; Ahn et al., 2023; Gatmiry

---

[*]Correspondence author

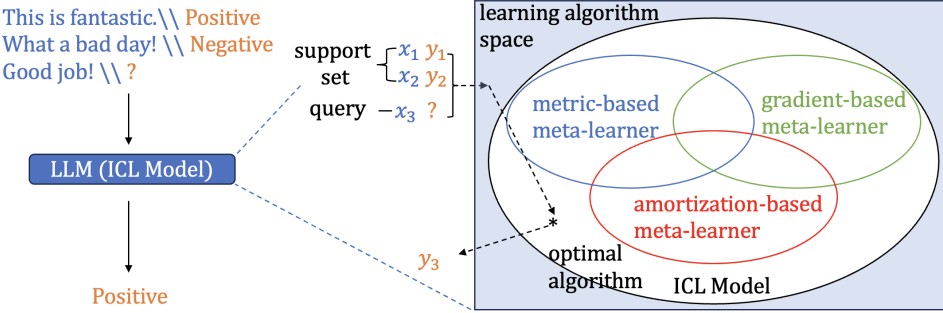

Figure 1: In-context learning models can find the data-dependent optimal learning algorithm, in a space which is more inclusive than typical meta-learners.

et al., 2024). Nevertheless, this setting is so simplified that it is far removed from real-world scenarios, and no more complex settings currently offer such a transparent understanding. To achieve a more generalizable understanding of ICL, researchers have approached the problem from various perspectives, including theoretical results on expressiveness (Wang et al., 2024b; Bai et al., 2023), learning dynamics and convergence (Tian et al., 2023; Li et al., 2023b; Huang et al., 2024; Zhang et al., 2024; Sander et al., 2024), generalization error (Li et al., 2023a; 2024; Wies et al., 2023), and observations of ICL model behaviors (Akyürek et al., 2023; Bhattamishra et al., 2024; Zhang et al., 2023).

Although precisely understanding what and how do ICL models learn from pre-training is challenging and depends on various problem settings and data distributions, a basic consensus has been reached. Specifically, it is understood that an ICL model learns a learning algorithm that maps $(\mathbf{x}^{(1)}, \mathbf{y}^{(1)}, \cdots, \mathbf{x}^{(n)}, \mathbf{y}^{(n)})$ to $f$ through pre-training. The inference process is then interpreted as first learning $f$ from $(\mathbf{x}^{(1)}, \mathbf{y}^{(1)}, \cdots, \mathbf{x}^{(n)}, \mathbf{y}^{(n)})$ and subsequently applying it to a new input $\mathbf{x}^{(n+1)}$. This consensus highlights the nature of ICL models as meta-learners (Kirsch et al., 2022; Dai et al., 2023), which involves learning a learning algorithm to enable systems to quickly adapt to new tasks—essentially, learning-to-learn (Schmidhuber, 1987; Thrun & Pratt, 1998). Given tasks for meta-training (pre-training), the goal is to learn a learner function (i.e., a learning algorithm) that can make inferences for a given input based on a provided set of labeled examples, enabling generalization to meta-testing (unseen) tasks. While typical meta-learners have been extensively studied, none have demonstrated the level of general intelligence achieved by LLMs, or ICL models. This naturally raises the question: what distinguishes ICL models from typical meta-learners? While existing works on understanding ICL focus on identifying the exact learning algorithms that ICL models learn, we aim to address a different question: *why are ICL models more prominent compared to typical meta-learners*?

The difference between ICL models and typical meta-learners lies in their hypothesis spaces. ICL is seen as meta-learning with minimal inductive bias (Kirsch et al., 2022). In machine learning, less inductive bias, often leads to better performance with sufficient data (LeCun et al., 2015). Human-designed knowledge may not always be relevant, while data-driven knowledge, optimized through training data, performs well when the hypothesis space is expressive and training is generalizable. Meta-learners' hypothesis spaces define the knowledge needed for a learning algorithm. They use a function with two inputs: a support set with labeled examples and a query input, producing the prediction. Typical meta-learners rely on strong prior knowledge to structure their algorithms. In contrast, ICL models use a black-box model, like transformers (Vaswani, 2017), which enable data-driven interactions among samples, can be stacked into deep architectures, and incorporate necessary inductive biases. These properties enhance their generalizability. We study if the advantage of data-driven approaches over human-designed knowledge contributes to the success of ICL models (Figure 1). We believe that ICL models are prominent due to their ability to learn data-dependent optimal learning algorithms. However, their optimality is data-dependent[1], posing potential risks in generalizability.

---

[1] The formal definitions of an optimal learning algorithm and a data-dependent optimal learning algorithm are provided in Appendix B

In this paper, we first prove the expressiveness of ICL with transformers as learning algorithms, and then we verify the aforementioned conjecture. Our findings indicate that ICL models are capable of learning data-dependent optimal algorithms, but their generalizability is limited. This limitation is evident from their distribution-sensitive performance when the algorithm is implicit, prompting us to reconsider the interpretation of ICL as 'algorithm selection' in existing literature. Building on this insight, we identify shared challenges between training deep models in supervised learning and pre-training ICL models. We then propose leveraging deep-learning techniques to enhance ICL by mapping supervised learning approaches to meta-learning. Our contributions are summarized as follows:

- We investigate ICL model from a learning-to-learn perspective by comparing it with typical meta-learners, understanding it as a meta-learner with minimal inductive bias.

- We theoretically prove that ICL with transformer is expressive enough to mimic existing meta-learners. We demonstrate its ability to construct data-dependent optimal algorithms across a wide range of settings. We also show that its generalizability is limited, as it implicitly fits the training distribution rather than learning an explicit algorithm.

- We proposed to transfer techniques in classical deep networks to improve ICL. As examples, we enhance the domain adaptability of ICL models through pre-training with meta-level meta-learning and accelerate convergence through pre-training with meta-level curriculum learning.

## 2 EXPRESSIVENESS OF ICL AS META-LEARNERS

### 2.1 A UNIFIED FORMULATION OF LEARNING-TO-LEARN

We first construct the formulation to align ICL models and typical meta-learners. To investigate learning algorithms as learnable functions, a learning algorithm (Kirsch et al., 2022) is considered as a mapping from a labeled dataset $\mathcal{D} = \{(\mathbf{x}^{(i)}, \mathbf{y}^{(i)})\}_{i=1}^n$ and a query input $\mathbf{x}^{(q)}$ to a prediction $\hat{\mathbf{y}}^{(q)}$. The function of a learning algorithm can all be represented as a learner function $g$:

$$\hat{\mathbf{y}}^{(q)} = g(\mathbf{x}^{(q)}, \mathcal{D}). \tag{1}$$

Learning-to-learn (Vilalta & Drissi, 2002; Hospedales et al., 2021), also known as meta-learning, aims to optimize a learnable function $g(\cdot; \theta)$ through meta-training. The process of training ICL models or other meta-learners exemplifies learning-to-learn.

**In-Context Learning with Transformer** Generally, there is a input matrix $Z_0$ composed of $\mathcal{D}$ and $\mathbf{x}^{(q)}$, which is fed into a $M$-layer transformer $\mathsf{TF}_M$. Denote the collection of all model weights in $\mathsf{TF}_M$ as $\theta_M$. ICL can thus be represented as a learner function $g_M$:

$$g_M(\mathbf{x}^{(q)}, \mathcal{D}; \theta_M) = \mathsf{TF}_M(Z_0; \theta_M), \tag{2}$$

with details on the construction of $Z_0$, the model architecture of $\mathsf{TF}_M$, and the optimization of $\theta_M$ provided in Appendix A.1.

**Meta-Learning Methods** Typical meta-learners are more restricted to certain learning algorithm structures designed by human experts. Strong inductive biases are introduced into $g(\cdot; \theta)$, defining how to adapt to $\mathcal{D}$ and make inference for $\mathbf{x}^{(q)}$. Typical meta-learners are generally classified into three categories (Bronskill et al., 2021): gradient-based, metric-based and amortization-based. Below is a summary of the function of each category.

- *Gradient-Based.* Given a prediction model $h : \mathbf{x} \mapsto \mathbf{y}$ and a loss function $\ell(\cdot, \cdot)$, gradient-based meta-learners (Finn et al., 2017) perform gradient-descent using the labeled data in $\mathcal{D}$:

$$g_{\text{gd}}(\mathbf{x}^{(q)}, \mathcal{D}; \theta) = h(\mathbf{x}^{(q)}; \theta - \sum\nolimits_{i=1}^n \nabla_\theta \ell(h(\mathbf{x}^{(i)}; \theta), \mathbf{y}^{(i)})). \tag{3}$$

- *Metric-Based.* Metric-based learners (Koch et al., 2015; Garcia & Bruna, 2018; Sung et al., 2018) learn to compare the query with examples by optimizing a distance metric in the feature space. Let

$d_\theta(\cdot, \cdot)$ denote a distance function. Pair-wise metric-based algorithm makes prediction based on the pair-wise distance between the query and examples:

$$g_{\text{sim}}(\mathbf{x}^{(q)}, \mathcal{D}; \theta) = \frac{1}{n} \sum\nolimits_{i=1}^{n} d_\theta(\mathbf{x}^{(i)}, \mathbf{x}^{(q)}) \mathbf{y}^{(i)}. \tag{4}$$

For classification tasks where $\mathbf{y}^{(i)} \in \{\boldsymbol{c}_c\}_{c=1}^{C}$, one can also adopt class-prototype metric-based algorithm (Snell et al., 2017), which compares the query with the class prototypes:

$$g_{\text{prt}}(\mathbf{x}^{(q)}, \mathcal{D}; \theta) = \sum\nolimits_{c=1}^{C} d_\theta(\frac{1}{C} \sum\nolimits_{y^{(i)}=c} \mathbf{x}^{(i)}, \mathbf{x}^{(q)}) \boldsymbol{c}_c, \tag{5}$$

where $\boldsymbol{c}_c$ is the class prototype of class $c$.

- *Amortization-Based.* Amortization-based meta-learners replace the optimization process with a single forward pass of the encoder to efficiently learn informative modulations (Garnelo et al., 2018; Ravi & Beatson, 2019). Typically, they follow a framework that uses a set encoder (Zaheer et al., 2017) to map $\mathcal{D}$ to a vector $\mathbf{e}$ representing the task context, then feeds the context $\mathbf{e}$ and the query to a prediction model $f_\theta : (\mathbf{x}, \mathbf{e}) \mapsto \mathbf{y}$. Considering the universal approximation property of neural networks, an amortization-based meta-learner can be formulated as:

$$g_{\text{am}}(\mathbf{x}^{(q)}, \mathcal{D}; \theta) = f_\theta(\mathbf{x}^{(q)}, \frac{1}{n} \sum\nolimits_{i=1}^{n} [\mathbf{x}^{(i)} | \mathbf{y}^{(i)}]). \tag{6}$$

The meta-training process is provided in Appendix A.2.

## 2.2 Understanding ICL through Unified Formulation

With the unified formulation, we study the expressiveness of ICL models compared with typical meta-learners. Expressiveness in deep-learning refers to a model's ability to capture complex patterns and relationships within data (LeCun et al., 2015), a fundamental property that enables deep learning models to achieve high performance on intricate tasks. Here, we focus on expressiveness at the meta-level—specifically, the ability to capture interaction patterns and relationships among samples in $\mathcal{D}$ and $\mathbf{x}^{(q)}$, which corresponds to the expressiveness of learning algorithms. In this section, we prove that **ICL with transformer is expressive enough to perform any learning algorithm that typical meta-learners can**.

Specifically, we show that, with certain parameter instantiations, ICL with transformer $g_M$ (2) can perform gradient-based $g_{\text{gd}}$ (3), pair-wise metric-based $g_{\text{sim}}$ (4), class-prototype metric-based $g_{\text{prt}}$ (5) and amortization-based $g_{\text{am}}$ (6). Since class-prototype metric-based methods are applicable only to classification tasks, we consider standard $C$-class classification tasks. Formally, we present the following theorems for classification problems where $C < \infty$:

**Theorem 2.1.** $\forall\ g\ \in\ \{g_{gd}, g_{sim}, g_{prt}, g_{am}\},\ \forall\ \theta\ \in\ \mathbb{R}^{|\theta|},\ \exists\ M\ \in\ \mathbb{N}^*\ <\ \infty,\ \exists\ \theta_M\ \in\ \mathbb{R}^{|\theta_M|},$ $g_M(\mathbf{x}^{(q)}, \mathcal{D}; \theta_M) = g(\mathbf{x}^{(q)}, \mathcal{D}; \theta).$

The detailed settings, the mild assumptions and proofs for this part, are provided in Appendix C.

## 3 ICL Model Does Learn Data-Dependent Optimal Algorithm

We have shown that the ICL model is expressive by proving that its hypothesis space is inclusive, at least covering the capabilities of typical meta-learners. However, the specific solution that the ICL model achieves within this space through pre-training on a given task set—i.e., the exact learning algorithm it learns—directly determines its performance and generalizability. Understanding this is crucial to investigating its prominence. In this section, we investigate whether the ICL model, with sufficient pre-training, learns an optimal learning algorithm from the training tasks, and examine its generalizability when the specific learning algorithm is not explicitly known.

**Algorithm Criterion.** To determine whether two learning algorithms are identical, particularly for classification tasks, their classification boundaries can be visualized through Monte Carlo sampling of query inputs. Given multiple trials with different sets of labeled examples, if two learner functions consistently produce the same classification boundary and exhibit identical end-to-end performance, we can infer with high probability that they represent the same learning algorithm.

**Generalizability of Learning Algorithm.** We use the terms **explicit** and **implicit** optimal learning algorithm to distinguish the generalizability of a learning algorithm. Formally, we define explicit optimal algorithm $g(\cdot; \mathcal{F}, *)$ of a function family $\mathcal{F}$ as: when $n \to \infty$, $\forall f \in \mathcal{F}$, $\forall p(x)$, $\mathbb{E}_{p(x)}[g(\mathbf{x}^{(q)}, \mathcal{D}; \mathcal{F}, *)] = f(\mathbf{x}^{(q)})$, where $\mathcal{D} = \{(\mathbf{x}^{(i)}, f(\mathbf{x}^{(i)}))\}_{i=1}^{n}$, $\mathbf{x}^{(q)} \sim p(x)$, $\mathbf{x}^{(i)} \sim p(x)$. In other words, An explicit optimal learning algorithm for $\mathcal{F}$ is generalizable across any data distribution $p(x)$, allowing it to learn any function $f \in \mathcal{F}$. In contrast, implicit optimal learning algorithms are sensitive to the specific data distribution. We denote $\boldsymbol{\mathcal{G}_\Omega}$ as the set of all ground-truth explicit optimal algorithms for a task set $\boldsymbol{\Omega}$. For example, Ordinary Least Squares is an explicit optimal learning algorithm $g(\cdot; \mathcal{F}, *)$ for linear regression tasks ($\mathcal{F} = \{f \mid f(\mathbf{x}) = \mathbf{w}^\top \mathbf{x}\}$), while memorizing and looking-up is an implicit optimal learning algorithm for any problem.

## 3.1 GENERATING TASKS WITH EXPLICIT OPTIMAL ALGORITHMS

To determine whether ICL with transformers learns the optimal learning algorithm, we generate tasks whose optimal predictions can be precisely achieved by certain explicit learning algorithms. For specific algorithms, we select representatives from each category of typical meta-learners: $g_{\text{sim}}$, $g_{\text{prt}}$ and $g_{\text{am}}$. Specifically, denoting a set of tasks $\boldsymbol{\Omega} = \{\mathcal{D}_\tau\}_{\tau=1}^{T}$, we generate three types of tasks: pair-wise metric-based tasks $\boldsymbol{\Omega}_{\text{sim}}$ where MatchNet (Vinyals et al., 2016) ($\in g_{\text{sim}}$) is the optimal learner, class-prototype metric-based tasks $\boldsymbol{\Omega}_{\text{prt}}$ where ProtoNet (Snell et al., 2017) ($\in g_{\text{prt}}$) is the optimal learner, amortization-based tasks $\boldsymbol{\Omega}_{\text{am}}$ where CNPs (Garnelo et al., 2018) ($\in g_{\text{am}}$) is the optimal learner. We do not consider $g_{\text{gd}}$ for two reasons: (i) it is challenging to define a family of classification tasks and the corresponding $h$ to guarantee the optimum; and (ii) proving that ICL can express $g_{\text{gd}}$ is not considered as a contribution of this paper, as it is straightforward by leveraging results from Bai et al. (2023); Wang et al. (2024b). Details regarding task generation and experimental settings are provided in Appendix D.

## 3.2 ICL MODEL LEARNS EXPLICIT OPTIMAL ALGORITHM ON SIMPLE TASKS

To verify that ICL with transformers learns the optimal algorithm, we perform meta-training using a single type of task corresponding to one explicit optimal algorithm. We then draw conclusions by comparing the classification boundaries of the trained ICL model with those of the known optimal algorithm. Additionally, we augment the above-mentioned optimal meta-learners with parameterized feed-forward layers, allowing them to be meta-trained alongside the ICL model. This ensures that their optimality becomes data-dependent, similar to the ICL model. We conduct this investigation using the three types of tasks described above.

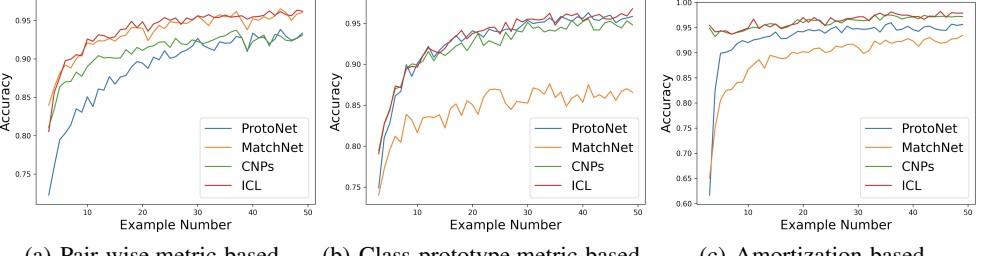

(a) Pair-wise metric-based.  (b) Class-prototype metric-based.  (c) Amortization-based.

Figure 2: Meta-testing performance of learners meta-trained and tested on the same single type of tasks.

Figure 2(a) shows the end-to-end performance of learners trained on $\boldsymbol{\Omega}_{\text{sim}}$ and tested on $\boldsymbol{\Omega}'_{\text{sim}}$(unseen tasks). The results indicate that the end-to-end performance of the ICL model is only marginally different from MatchNet, the parameterized optimal meta-learner obtained through meta-training. To further verify this, we visualize how the ICL model classifies each sample given a few fixed labeled examples. An example is shown in the first column in Figure 5, with additional cases provided in Appendix E.1. These visualizations confirm that the ICL model learns the same algorithm as MatchNet, which is data-dependent and optimal for $\boldsymbol{\Omega}_{\text{sim}}$. Similarly, we observe that the ICL model learns ProtoNet on $\boldsymbol{\Omega}_{\text{prt}}$, where ProtoNet is the optimal learner for this task type, as shown in Figure 2(b), and the second column in Figure 5. Likewise, the ICL model learns CNPs on $\boldsymbol{\Omega}_{\text{am}}$, the optimal

learner for these tasks, as demonstrated in Figure 2(c), and the third column in Figure 5. Results of more trials are provided in Appendix E.1. Thus, we conclude that when pre-trained on $\Omega_{\text{sim}}$, $\Omega_{\text{prt}}$ or $\Omega_{\text{am}}$, **the ICL model learns the explicit optimal algorithm**.

### 3.3 ICL Model Learns Implicit Optimal Algorithm on Mixed Tasks

In real-world scenarios, tasks are often complex and diverse. Therefore, we further investigate what learning algorithm the ICL model learns when the pre-training tasks come from various types that do not share a single optimal explicit learning algorithm.

Specifically, we mix the above $\Omega_{\text{sim}}$, $\Omega_{\text{prt}}$ and $\Omega_{\text{am}}$ to form meta-training task set $\Omega_{\text{mix}}$, such that $\mathcal{G}_{\Omega_{\text{mix}}} = \{\text{MatchNet}, \text{ProtoNet}, \text{CNPs}\}$. We meta-train ICL model, MatchNet, ProtoNet and CNPs with $\Omega_{\text{mix}}$, and evaluate their performance on unseen $\Omega'_{\text{sim}}$, $\Omega'_{\text{prt}}$ and $\Omega'_{\text{am}}$ respectively. Figure 3 shows the results. We also compare with the performance of data-dependent optimal algorithm (D.-Dpt. Optimal) for each type of testing tasks. In Figure 3(a), D.-Dpt. Optimal refers to MatchNet trained on $\Omega_{\text{sim}}$ and tested on $\Omega'_{\text{sim}}$; in Figure 3(b), it refers to ProtoNet trained on $\Omega_{\text{prt}}$ and tested on $\Omega'_{\text{prt}}$; and in Figure 3(c), it refers to CNPs trained on $\Omega_{\text{am}}$ and tested on $\Omega'_{\text{am}}$. We also visualize the classification boundaries of the ICL model on testing tasks, which align with those of the optimal algorithms. Examples of these patterns are shown in Figures 10–12 and are further detailed in Appendix E.2.

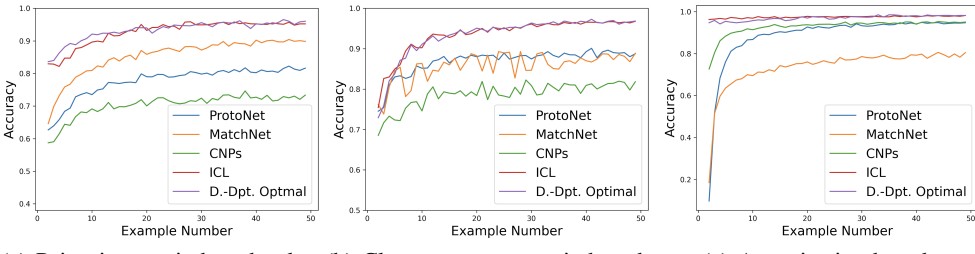

(a) Pair-wise metric-based tasks. (b) Class-prototype metric-based. (c) Amortization-based.

Figure 3: Meta-testing performance of learners meta-trained on hybrid tasks, tested on each seen task type separately.

From the above results, we draw several conclusions. First, typical meta-learners trained on $\Omega_{\text{mix}}$ (MatchNet, ProtoNet, CNPs) fail to achieve the data-dependent optimal (D.-Dpt. Optimal) performance on $\Omega'_{\text{sim}}$, $\Omega'_{\text{prt}}$ or $\Omega'_{\text{am}}$, indicating that none of thses explicit learning algorithm can simultaneously be optimal for $\Omega_{\text{sim}}$, $\Omega_{\text{prt}}$ and $\Omega_{\text{am}}$. Second, the ICL model trained on $\Omega_{\text{mix}}$ (ICL) demonstrates performance very similar to data-dependent optimal (D.-Dpt. Optimal) models. Furthermore, the identical classification boundaries presented in Appendix E.2 suggest that the ICL model successfully learns a data-dependent optimal learning algorithm on $\Omega_{\text{mix}}$. Finally, these findings collectively demonstrate that the ICL model is more expressive than typical meta-learners and is capable of learning a data-dependent optimal algorithm that $\notin \mathcal{G}_{\Omega_{\text{mix}}}$. However, the precise nature of this learned algorithm remains unknown.

We concern about the question of whether the data-dependent optimal learning algorithm on $\Omega_{\text{mix}}$ is implicit or explicit, as this is directly related to the generalizability of the ICL model. Existing works have studied ICL models trained on mixed types of tasks to perform "algorithm selection" (Li et al., 2023a; Bai et al., 2023; Bhattamishra et al., 2024; Wang et al., 2024b). "Algorithm selection" refers to the process where, among all algorithms that are explicitly optimal for specific pre-training tasks, the ICL model selects the most suitable algorithm based on the specific task context. This can be formally described when trained with $\Omega_{\text{mix}}$:

$$g_M(\mathbf{x}^{(q)}, \mathcal{D}; \theta_M) = g^*(\mathbf{x}^{(q)}, \mathcal{D}), \tag{7}$$

$$s.t., \ g^* = \arg\min_{g \in \mathcal{G}_{\Omega_{\text{mix}}}} \sum_{\mathcal{D}' \subset \mathcal{D}} \sum_{(\mathbf{x}^{(i)}, \mathbf{y}^{(i)}) \in \mathcal{D}'} \ell(g(\mathbf{x}^{(i)}, \mathcal{D}/\mathcal{D}'), \mathbf{y}^{(i)}),$$

which is an end-to-end explicit optimal algorithm.

However, we question whether the ICL model truly learns the "algorithm selection" algorithm when trained on mixed types of tasks. While this interpretation may appear reasonable based on existing

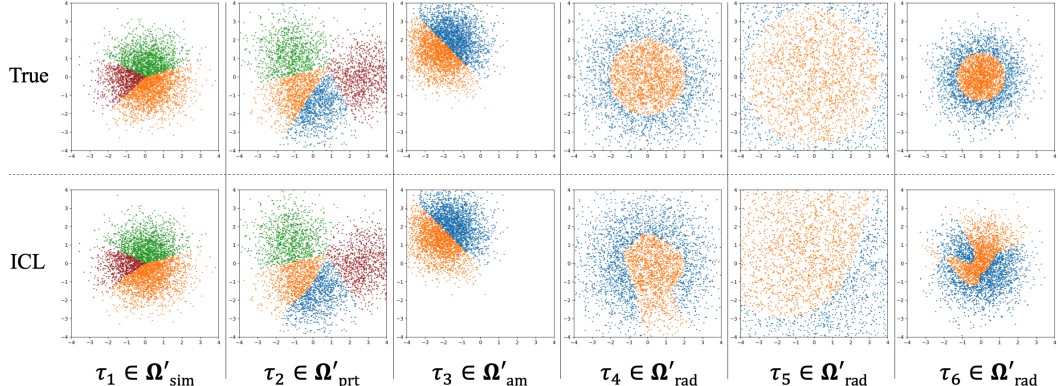

Figure 5: Comparing ICL's predictions and true labels with different types of tasks. Upper: true label. Lower: ICL prediction.

empirical results, it raises concerns about the model's generalizability to unseen task types and out-of-distribution data—an aspect that has not been thoroughly investigated in the literature. First, If ICL model trained with $\Omega_{\text{mix}}$ strictly follows (7), it would struggle catastrophically with tasks from novel types that cannot be solved by any $g \in \mathcal{G}_{\Omega_{\text{mix}}}$. Second, if the model learns an explicit optimal algorithm for tasks from seen types, it would lack the distribution sensitivity necessary to consistently handle tasks from seen types with data from varying distributions. The following results show that **ICL is not "algorithm selection"**.

### 3.3.1   ICL CAN SOLVE TASKS FROM UNSEEN TYPE

We continue to consider the ICL model trained on the above $\Omega_{\text{mix}}$, but now introduce a novel type of tasks for testing: radial distance tasks $\Omega_{\text{rad}}$. A task is generated by first sampling $r \in \mathbb{R}$ from $p(\tau)$, which defines the classification boundary. Then, $\{\mathbf{x}^{(i)} \in \mathbb{R}^d\}_{i=1}^N$ are sampled from $p(x)$, and labels are assigned as follows: $\mathbf{y}^{(i)} = 0$ if $||\mathbf{x}^{(i)}|| \leq r$ otherwise $\mathbf{y}^{(i)} = 1$. Note that we specifically control $p(x)$ based on $r$ to ensure that the labels are balanced within each task.

Such radial distance tasks cannot be solved by following the "algorithm selection" approach, as a radial distance task $\mathcal{D}$ cannot be learned by any $g \in \mathcal{G}_{\Omega_{\text{mix}}} = \{\text{MatchNet}, \text{ProtoNet}, \text{CNPs}\}$. Formally, $\forall g \in \mathcal{G}_{\Omega_{\text{mix}}}, \mathbb{E}_{p(\tau),p(x)}[\sum_{(\mathbf{x}^{(i)},\mathbf{y}^{(i)}) \in \mathcal{D}} \ell(g(\mathbf{x}^{(i)}, \mathcal{D}/(\mathbf{x}^{(i)}, \mathbf{y}^{(i)})), \mathbf{y}^{(i)})] = l$, where $l$ represents the expected loss of making predictions through random guessing. Following the "algorithm selection" interpretation in (7), no $g^*$ can be determined. Even if an arbitrary $g^*$ were selected and used, its performance would not improve with an increasing number of examples.

However, we find that ICL model trained with $\Omega_{\text{mix}}$ can handle radials distance tasks effectively. While the exemplar tasks in the 4-6th columns in Figure 5 show that ICL model does not solve them optimally, the increasing accuracy with a growing number of examples, as shown in Figure 4(a), indicates that it effectively learns task-specific information—something that an "algorithm selector" cannot achieve. We conjecture that when $g \in \Omega_g$ is inclusive, the pre-trained ICL could be generalized to diverse tasks, including those from novel types. This property likely contributes to the success of LLMs by approaching the ideal of "learning-to-learn."

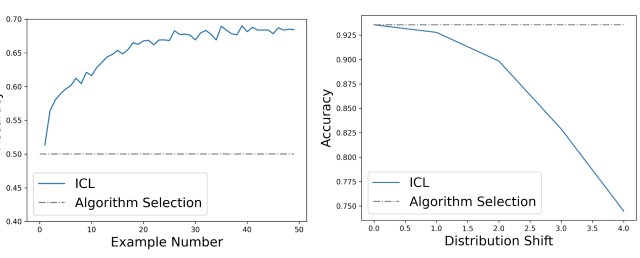

(a) Radial distance tasks.    (b) Under distribution shift.

Figure 4: Performance comparing between ICL and ideal "algorithm selection".

### 3.3.2 ICL Shows Distribution-Sensitive Generalizability

Another result reveals that the ICL model has limited generalizability, even on tasks from seen types, as it is sensitive to the data distribution. In contrast, an "algorithm selector," as an explicit optimal algorithm, would not exhibit such sensitivity. Figure 4(b) shows the ICL's performance on hybrid tasks from seen type. The ICL model is trained with $\mathbf{\Omega}_{\text{mix}}$ from with input distribution $p(x)$, but tested on $\mathbf{\Omega}'_{\text{mix}}$ with a shifted distribution $p'(x)$. The performance decreases noticeably as the distribution shift between training and testing (x-axis) increases, despite the tasks being from seen types. This result directly addresses our question, demonstrating that **though ICL model trained with $\mathbf{\Omega}_{\text{mix}}$ does learn a data-dependent optimal learning algorithm, it is implicit**. Consequently, the ICL model exhibits limited generalizability, as its meta-testing performance is sensitive to data distribution. This behavior reflects the generalizability characteristics of deep learning, now observed at the meta-level.

## 4 Improving ICL Models by Transferring Techniques for Classical Deep Networks

Above, we have seen that ICL model can be seen as a data-dependent optimal meta-learner. However, the learned data-dependent optimal algorithm is fitting the training distribution, limiting generalizability — a behavior similar to well-known characteristics of classical deep networks. There are many techniques to enhance classical deep networks' performance, e.g., contrastive learning (Khosla et al., 2020), meta-learning (Finn et al., 2017), curriculum learning (Bengio et al., 2009), and data augmentation (Shorten & Khoshgoftaar, 2019). We are motivated to transfer exemplar techniques that are originally for deep networks to address the above problems to ICL here.

From a motivational perspective, classical deep networks and ICL models share fundamental challenges. Both require inclusive training data for generalization, but training data is often limited in real-world. They also share common goals, such as improving generalizability, accelerating convergence, and enabling fast adaptation. Thus techniques designed to address these issues in classical deep networks may also be effective for ICL models. From a technical perspective, many techniques designed to achieve the above goals do not impose strict analytical restrictions on the loss function or model architecture, requiring only a differentiable supervision signal. And the inference process of ICL also only relies on feeding-forward. Therefore, these techniques can at least be implemented at the meta-level through the mapping from samples to tasks, sample-wise loss to task-wise auto-regressive loss, and epochs to episodes. Here, we discuss two exemplary practices: meta-level meta-learning, which significantly improves ICL performance on specific domains with very limited data for adaptation, and meta-level curriculum learning, which effectively accelerates the pre-training process.

### 4.1 Meta-Level Meta-learning

While general ICL models can be applied directly, adapting them to domain-specific data is common for building domain-specific intelligence. However, limited domain-specific data increases the risk of overfitting, a few-task problem akin to the few-shot issue in supervised learning (Wang et al., 2020), illustrated in detail in Appendix G. Techniques like LoRA (Hu et al., 2022) and prefix-tuning (Li & Liang, 2021) address this by efficiently adapting ICL models. Unlike existing methods that assume a pre-trained general ICL model pre-trained by (13) is given, we consider pre-training specifically for adaptation, optimizing performance after few-task domain adaptation. This meta-level meta-learning approach mimics few-task adaptation during pre-training by solving a bi-level optimization problem. It is practical as real-world pre-training tasks are often naturally divided into semantic domains.

Consider a domain distribution $p(\delta)$. Each domain $\delta$ determines a distribution of tasks $p_\delta(\tau)$ where a domain-specific task set $\mathbf{\Omega}_\delta = \{\mathcal{D}_\tau\}_{\tau=1}^{T_\delta}$ can be drawn. During pre-training, we manually split $\mathbf{\Omega}_\delta$ into two disjoint task sets: a training (support) task set $\mathbf{\Omega}_\delta^{\text{tr}} = \{\mathcal{D}_\tau\}_{\tau=1}^{t}$ and a validation (query) task set $\mathbf{\Omega}_\delta^{\text{val}} = \{\mathcal{D}_\tau\}_{\tau=t+1}^{T_\delta}$. Denote a meta-level meta-learner as $G(g, \mathbf{\Omega}; \Delta)$, i.e., a domain adapter adapting meta-learner $g$ with task set $\mathbf{\Omega}$. Denote a meta loss function evaluating meta-learner $g$ with $\{\mathcal{D}_\tau^{\text{tr}}, \mathcal{D}_\tau^{\text{val}}\}$ as $\ell_{\text{meta}}(\tau, g)$. Meta-level meta-training is performing:

$$\min_\Delta \mathbb{E}_{p(\delta)} \left[ \frac{1}{|\mathbf{\Omega}_\delta^{\text{val}}|} \sum_{\tau \in \mathbf{\Omega}_\delta^{\text{val}}} \ell_{\text{meta}}(\tau, G(g(\cdot; \theta), \mathbf{\Omega}_\delta^{\text{tr}}; \Delta)) \right]. \tag{8}$$

Specifically, to adopt meta-level meta-learning for improving the pre-training of the ICL model $g(\cdot;\theta) = g_M(;\theta_M)$, we implement MAML as the meta-level meta-learner: $G(g(\cdot;\theta), \boldsymbol{\Omega}_\delta^{\text{tr}}; \Delta) = g(\cdot; \theta - \nabla_\theta \frac{1}{|\boldsymbol{\Omega}_\delta^{\text{tr}}|} \sum_{\tau \in \boldsymbol{\Omega}_\delta^{\text{tr}}} \ell_{\text{meta}}(\tau, g(\cdot;\theta)))$, which avoids designing additional learnable $\Delta$ due to MAML's model-agnostic property. Alternatively, one can also design meta-level meta-learners based on metric or amortization-based approaches.

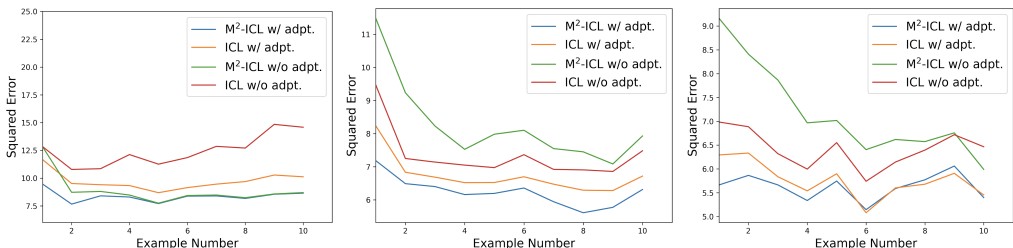

(a) Given 64 tasks for adaptation. (b) Given 256 tasks for adaptation. (c) Given 1024 tasks for adaptation.

Figure 6: Performance of meta-trained ICL model and meta-level meta-trained ICL model on unseen domain, given a few tasks for adaptation.

We conduct experiments on linear regression tasks, where the distribution of linear weights is $p_\delta(\tau) = \mathcal{N}(\mu_\delta, \Sigma_\delta)$ with $(\mu_\delta, \Sigma_\delta) \sim p(\delta)$. More details are provided in Appendix F. We denote such meta-level meta-trained ICL model as $\mathbf{M^2\text{-}ICL}$. After pre-training, we test on unseen domains drawn from $p(\delta)$. Each domain provides $\boldsymbol{\Omega}_\delta^{\text{tr}} = \{\mathcal{D}_\tau\}_{\tau=1}^{t}$ for adaptation, and $\boldsymbol{\Omega}_\delta^{\text{val}} = \{\mathcal{D}_\tau\}_{\tau=t+1}^{T_\delta}$ for performance evaluation. The performance is shown in Figure 6. Note that reasonable solutions include ICL w/ adpt, ICL w/o adpt, and $\text{M}^2$-ICl w/ adpt, while $\text{M}^2$-ICl w/o adpt serves only as an intermediate product of meta-level meta-learning. We find that $\text{M}^2$-ICL w/ adpt outperforms both ICL w/ adpt and ICL w/o adpt, particularly when the number of adaptation tasks is very small (64, Figure 6(a)), while the advantage gradually decreases with the growth of task number (marginal with 1024 adaptation tasks Figure 6(c)). Meta-level meta-learning is effective for fast adaptation on few-task domain, like typical meta-learning's effectiveness for fast adaptation on few-shot task. Note that, although the adaptation strategy in this experiment involves fine-tuning all parameters using gradient descent (i.e., $G$ is derived from MAML with inner updates as full-parameter fine-tuning), any differentiable adaptation strategy can replace the inner-update or be incorporated into other specifications of $G$. The comparison between ICL and $\text{M}^2$-ICL is isomorphic with the comparison between a model trained using standard supervised learning and a model meta-trained using MAML. Experiment results on cross-domain few-shot image classification is provided in Appendix H.

## 4.2 META-LEVEL CURRICULUM LEARNING

Curriculum meta-learning strategy is intuitive, as a meta-learner should progressively learn tasks from simple to complex for better convergence (Bengio et al., 2009). This approach has been shown to be effective for gradient-based meta-learners (Chen et al., 2021; Stergiadis et al., 2021). Here, we investigate whether the curriculum strategy can enhance the meta-training of ICL.

We consider a simple case: an ICL model learning linear regression tasks, where task complexity is evaluated by the number of dimensions. This approach, practiced by Garg et al. (2022), has not yet been explicitly investigated for its effect. With a maximum dimension of 20, we train the ICL model on tasks with an increasing number of effective dimensions (denoted such baseline as **CL-ICL**, which curriculum is shown in Figure 7(a)), while

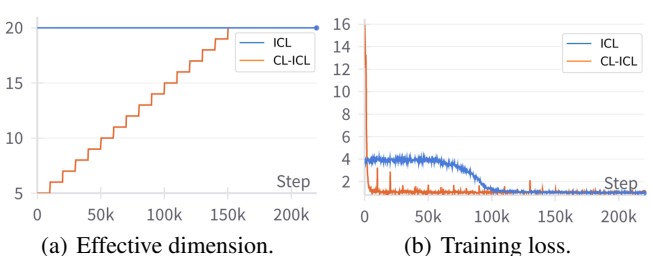

(a) Effective dimension. (b) Training loss.

Figure 7: Training dynamics of ICL model with curriculum.

values in the remaining dimensions are set to zero. The training loss (Figure 7(b)) and performance

comparison under limited training (Figure 8(a)) demonstrate that this curriculum enables faster convergence. However, with sufficient training, the ICL models trained with and without the curriculum exhibit nearly identical training loss and testing performance (Figure 8(b)). This result suggests that, in this case, the curriculum strategy accelerates convergence but does not lead to a better optimum.

## 5 RELATED WORKS

Here, we provide a detailed discussion of the connections between this paper and the most relevant studies. It has been comprehensively investigated (Akyürek et al., 2023; Bai et al., 2023) that the explicit learning algorithms that transformers can learn in-context, including ridge regression, least squares, and Lasso on linear regression tasks. Building on their findings, we believe that transformers are capable of learning numerous explicit learning algo-

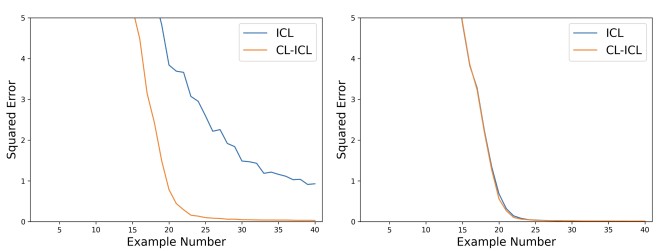

(a) Testing @ $2 \times 10^5$ episodes.  (b) Testing @ $5 \times 10^5$ episodes.

Figure 8: Testing performance of ICL model with curriculum.

rithms in-context. Rather than focusing on identifying specific explicit algorithms under different settings, our work provides a broader and more abstract understanding of ICL with transformers. We conceptualize ICL as a data-dependent model that can express typical meta-learners. These meta-learners are capable of expressing a wide range of explicit learning algorithms (Finn & Levine, 2018; Zaheer et al., 2017) and can be applied to a variety of broad applications (Wu et al., 2024; Yao et al., 2024; Wang et al., 2024a). It has also been proven that ICL with transformers can implement gradient descent on neural networks (Bai et al., 2023), which we leverage as an important tool to show that ICL can perform gradient-based meta-learning algorithms.

We adopt the definition of a learning algorithm from Kirsch et al. (2022) and build on their understanding that ICL models function as general-purpose meta-learning systems with minimal inductive bias. They demonstrate that ICL models can do learning-to-learn and a large number of training tasks matters. We compare ICL models with other meta-learners, proving the expressiveness of ICL with transformers and revealing its learnability and the characteristics of the algorithms it learns.

## 6 CONCLUSION, LIMITATIONS AND DISCUSSION

Pre-training an ICL model is essentially learning-to-learn. This paper compares ICL models with typical meta-learners, suggesting strategies to enhance ICL. It shows that ICL with transformers can learn optimal algorithms in a data-dependent way, but generalizability remains an issue as the learned algorithm may fit the training distribution rather than generalize well. ICL models in meta-learning are similar to deep models in supervised learning, sharing characteristics like data-dependent optimality and generalizability challenges. We propose applying deep-learning techniques, such as meta-level meta-learning and curriculum learning, to improve ICL. These strategies could enhance domain adaptability and speed up convergence, providing insights into ICL and a foundation for more robust models.

This paper examines ICL with transformers without considering the order of examples. Future research could explore the impact of positional embeddings in transformers, alternative architectures like RNNs and SSMs (Gu & Dao, 2023), and the convergence dynamics of pre-training ICL models, especially for real-world LLMs. Additionally, advanced techniques like contrastive learning and denoising could be adapted to the meta-level to further improve ICL performance.

ACKNOWLEDGEMENT

Q. Yao's work is supported by National Natural Science Foundation of China (under Grant No. 92270106), Beijing Natural Science Foundation (under Grant No. 4242039), and CCF-Huawei Populus Grove Fund.

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

## A PRELIMINARIES

### A.1 IN-CONTEXT LEARNING WITH TRANSFORMER

**Input.** Following existing works (Von Oswald et al., 2023; Ahn et al., 2023), we investigate ICL without positional embedding to study its learning-to-learn ability while ignoring the order of examples. Let $\mathbf{x}^{(i)} \in \mathbb{R}^d$ be a input, and $\mathbf{y}^{(i)} \in \mathbb{R}^e$ be the corresponding output. For each task $\tau$, there a task-specific function $f_\tau$ and dataset of the task $\mathcal{D}_\tau$ that $\forall (\mathbf{x}^{(i)}, \mathbf{y}^{(i)}) \in \mathcal{D}_\tau, \mathbf{y}^{(i)} = f_\tau(\mathbf{x}^{(i)})$.

Labeled examples and query of a task are input together. Define the input matrix $Z_0$:

$$Z_0 = \begin{bmatrix} \mathbf{z}^{(1)}\ \mathbf{z}^{(2)}\ \cdots\ \mathbf{z}^{(n)}\ \mathbf{z}^{(n+1)} \end{bmatrix} = \begin{bmatrix} \mathbf{x}^{(1)} & \mathbf{x}^{(2)} & \cdots & \mathbf{x}^{(n)} & \mathbf{x}^{(n+1)} \\ \mathbf{y}^{(1)} & \mathbf{y}^{(2)} & \cdots & \mathbf{y}^{(n)} & \boldsymbol{q} \end{bmatrix} \in \mathbb{R}^{(d+e) \times (n+1)}, \quad (9)$$

where $\boldsymbol{q} \in\in \mathbb{R}^e$ is the indicator of unlabeled query. ICL model is trained to output the prediction of $\mathbf{y}^{(n+1)}$ given $Z_0$, with a set of tasks $\{\mathcal{D}_\tau\}_{\tau=1}^T = \{\{(\mathbf{x}^{(i)}, \mathbf{y}^{(i)})\}_{i=1}^{N_\tau}\}_{\tau=1}^T$ from training distribution $f_\tau \sim p(\tau), \mathbf{x}^{(i)} \sim p(x)$, to generalize to unseen tasks.

**Model Architecture.** ICL is typically achieved by transformer, which are composed of stacked self-attention layers. Given $Z \in \mathbb{R}^{(d+e) \times (n+1)}$, a single-head self-attention layer $\mathrm{Attn}^{\mathsf{smax}}$ is defined as

$$\mathrm{Attn}^{\mathsf{smax}}_{W_{k,q,v}}(Z) = W_v Z \cdot \mathsf{smax}(Z^\top W_k^\top W_q Z), \quad (10)$$

where $W_v, W_k, W_q \in \mathbb{R}^{(d+e) \times (d+e)}$ are the (value, key and query) weight matrices, and $\mathsf{smax}(\cdot)$ is the softmax operator which applies softmax operation to each column of the input matrix. Note that the prompt is asymmetric since the label for $x^{(n+1)}$ is excluded from the input.

An $M$-layer transformer, denoted as $\mathsf{TF}_M$, consists of a stack of $M$ self-attention layers and MLP blocks. Formally, denoting by $Z_l$ the output of the $l^{\mathrm{th}}$ layer attention, we define

$$Z_{l+1} = Z_l + \sigma_{\alpha_l}(\mathrm{Attn}_{P_l, Q_l}(Z_l)) \quad \text{for } l = 0, 1, \ldots, M-1, \quad (11)$$

where $\sigma(\cdot)$ represents feed-forward layers parameterized by $\alpha_l$, which operates independently on each column of the input. Given $Z_0$, the prediction is obtained as

$$\hat{\mathbf{y}}^{(n+1)} = \mathsf{TF}_M(Z_0; \{P_l, Q_l, \alpha_l\}_{l=0}^{M-1}, W_r) = W_r[Z_M]_{:,(n+1)}, \quad (12)$$

where $[Z_M]_{:,(n+1)}$ is the $(n+1)$-th column of $Z_M$, and $W_r \in \mathbb{R}^{e \times (d+e)}$ is the linear readout weight.

During training, given the distribution of tasks $f_\tau \sim p(\tau), \mathbf{x}^{(i)} \sim p(x)$, and loss function $\ell(\cdot, \cdot)$ (e.g., cross-entropy), the parameters are optimized to minimize the expectation of auto-regressive loss of training tasks:

$$\min_{\{P_l, Q_l, \theta_l\}_{l=0}^{M-1}, W_r} \mathbb{E}_{p(\tau), p(x)} \Big[ \frac{1}{N_\tau} \sum_{i=0}^{N_\tau - 1} \ell(\hat{\mathbf{y}}^{(i+1)}, \mathbf{y}^{(i+1)}) \Big]. \quad (13)$$

### A.2 META-LEARNING

Meta-learning is a methodology concerned with "learning-to-learn" algorithms. Define $g(\cdot; \theta)$ is a meta-learner that maps a task dataset $\mathcal{D}_\tau$ and a query input $\mathbf{x}^{(q)}$ to its task-specific prediction. Typical meta-learning algorithms first learn an explicit model to a model $h$ from $\mathcal{D}_\tau$ and then perform the prediction, i.e, $\hat{\mathbf{y}}^{(q)} = g(\mathbf{x}^{(q)}, \mathcal{D}; \theta) = g'(\mathcal{D}; \theta)(\mathbf{x}^{(q)})$. For meta-training, given the training distribution $p(\tau)$ and $p(x)$, from which the tasks $\{\mathcal{D}_\tau\}_{\tau=1}^T$ are drawn, the goal is to learn $g(\cdot; \theta)$ that performs well on unseen tasks. Within each task, a training set $\mathcal{D}_\tau^{\mathrm{tr}} = \{(x_{\tau,i}, y_{\tau,i})\}_{i=1}^n$ is used to provide supervised information, and a validation set $\mathcal{D}_\tau^{\mathrm{val}} = \{(x_{\tau,i}, y_{\tau,i})\}_{i=n+1}^{N_\tau}$ is used to evaluate performance and optimize the meta-learner. The meta-training process is then performed as:

$$\min_\theta \mathbb{E}_{p(\tau), p(x)} \Big[ \frac{1}{N_\tau - n} \sum_{i=n+1}^{N_\tau} \ell(g(\mathbf{x}^{(i)}, \mathcal{D}_\tau^{\mathrm{tr}}; \theta), \mathbf{y}^{(i)}) \Big]. \quad (14)$$

# B OPTIMALITY OF A LEARNING ALGORITHM

We claim ICL model learns data-dependent optimal learning algorithms (DDOLA), which is different and weaker than (true) optimal learning algorithm (OLA).

Formally, given a finite training set $\mathcal{D}_{\text{train}} = \{(x_i, y_i)\}$ where each sample is i.i.d.: $(x_i, y_i) \sim p(x, y)$, and a unseen testing set $\mathcal{D}_{\text{test}} = \{(x_j, y_j)\}$ following the same distribution, the OLA is $g^* = \arg\max_g \mathbb{E}_{(x_j, y_j) \sim p(x,y)}\{\text{Prob}[g(x_j, \mathcal{D}_{\text{train}}) = y_j]\}$. Which is to say a learning algorithm can make the most "accurate" prediction given a training set and unseen target input from the same distribution. It is possible to know the optimal learning algorithm with the priori of $p(x, y)$. For example, ordinary least squares is optimal for linear regression with Gaussian noise. In the paper, three types of tasks are generated by designed ways, i.e., known $p(x, y)$ (Section 3.1). It is obvious that a MatchNet model with certain parameters (simply keeping all modules inside as identical mappings) is the optimal learning algorithm for $\mathbf{\Omega}_{\text{sim}}$, and so does ProtoNet for $\mathbf{\Omega}_{\text{prt}}$ and CNPs for $\mathbf{\Omega}_{\text{am}}$. However, meta-learners have not access the true $p(x, y)$. They only learn the function to infer $p(x, y)$ from $\mathcal{D}_{\text{train}} = \{(x_i, y_i)\}$ through meta-training, which inevitably brings variance and bias, being (meta-training) data-dependent. So we denote that given certain meta-training set, the best that a random-initialized and meta-trained deep learner can do as the DDOLA. This could be empirically approximated by meta-training a deep and random-initialized MatchNet/ProtoNet/CNPs with certain meta-training set (for the three task types respectively).

# C PROOF OF THE META-LEVEL EXPRESSIVENESS OF ICL

**Theorem C.1.** $\forall\, g \in \{g_{gd}, g_{sim}, g_{prt}, g_{am}\}$, $\forall\, \theta \in \mathbb{R}^{|\theta|}$, $\exists\, M \in \mathbb{N}^* < \infty$, $\exists\, \theta_M \in \mathbb{R}^{|\theta_M|}$, $g_M(\mathbf{x}^{(q)}, \mathcal{D}; \theta_M) = g(\mathbf{x}^{(q)}, \mathcal{D}; \theta)$.

**Proof Sketch** The proof of is constructed by decomposing the functions of typical learners into $M \in \mathbb{N}^* < \infty$ conditioned steps, where each step can be achieved through one transformer layer with the following two basic tools:

1. Universal approximation property of multi-layer perceptron (MLP) (Hornik et al., 1989): This property allows feed-forward layers to express a wide range of functions $\mathbb{R}^{\dim_1} \to \mathbb{R}^{\dim_2}$. In each transformer layer, the feed-forward module operates independently on each column of the input matrix. This enables the transformer to perform a wide range of sample-wise transformations at every layer.
2. Orthonormal label embedding in $\mathbb{R}^{2C}$: We use a set of orthonormal vectors in $\mathbb{R}^{2C}$ as embeddings for the categorical labels (including the query identifier), such as one-hot embeddings. This ensures that the attention weight matrix $A \in \mathbb{R}^{(n+1) \times (n+1)}$ in the self-attention module of each transformer layer can be *label-aware*. Label-awareness implies that $\{A \in \mathbb{R}^{(n+1) \times (n+1)} \mid (\mathbf{y}^{(i)} = \mathbf{y}^{(i')}) \wedge (\mathbf{y}^{(j)} = \mathbf{y}^{(j')}) \Rightarrow A_{i,j} = A_{i',j'}\}$, i.e., the interaction weight between ordered a sample pair $(i, j)$ depends only on their labels $(\mathbf{y}^{(i)}, \mathbf{y}^{(j)})$, and can take any value in $\mathbb{R}^*$. This allows the learning algorithm to achieve the behavior of a label-aware set function.

The decomposition into finite conditioned steps is specific to each theorem and is not necessarily unique. Next we provide the full proof.

## C.1 DETAIL SETTINGS

### C.1.1 CLASSIFICATION TASK WITH ORTHONORMAL LABEL EMBEDDING

Classification task specifies the ICL's input in Section A.1 with $\mathbf{y}^{(i)} \in \{\boldsymbol{c}_1, \boldsymbol{c}_2, \cdots, \boldsymbol{c}_C\}$, where $\boldsymbol{c}_c$ is the embedding vector of the $c$-th class. For these label embeddings, we can find $2C$ orthonormal vectors in $\mathbb{R}^{2C}$: $\{\boldsymbol{u}_j\}_{j=1}^{2C}$, such that:

$$\boldsymbol{u}_i^\top \boldsymbol{u}_j = \begin{cases} 1, & \text{if } i = j \\ 0, & \text{if } i \neq j \end{cases}. \tag{15}$$

A simple choice of $\{\boldsymbol{u}_j\}_{j=1}^{2C}$ is the set of $2C$ one-hot vectors. We use $\{\boldsymbol{u}_j\}_{j=1}^{C}$ as the embeddings of $\{\boldsymbol{c}_c\}_{c=1}^{C}$, i.e., $\mathbf{y}^{(i)} \in \{\boldsymbol{u}_j\}_{j=1}^{C}$, and $\boldsymbol{u}_{C+1}$ as the indicator of query $\boldsymbol{q}$.

### C.1.2 SELF-ATTENTION WITHOUT SOFTMAX

In our setting, we consider self-attention layers that replace the softmax operation in (10) with column-wise $L1$-normalization. In particular, (10) is now approximated and reparameterized with weights $P := W_v \in \mathbb{R}^{(d+e)\times(d+e)}$ and $Q := W_k^\top W_q \in \mathbb{R}^{(d+e)\times(d+e)}$ as:

$$\text{Attn}_{P,Q}(Z) = PZ\,\text{norm}_1^{\text{col}}(Z^\top Q Z).\tag{16}$$

where $[\text{norm}_1^{\text{col}}(A)]_{i,j} = \begin{cases} \dfrac{A_{i,j}}{\sum_i |A_{i,j}|}, & \sum_i |A_{i,j}| \neq 0 \\ 0, & \sum_i |A_{i,j}| = 0 \end{cases}$ . Note that it is conventional to omit certain

non-linearities, such as the softmax operation, in self-attention layers to align transformers with explicit learning algorithms. While existing works often replace the softmax operation with $\frac{1}{n}$ (Von Oswald et al., 2023; Ahn et al., 2023), the $\text{norm}_1^{\text{col}}$ in (16) provides a closer approximation.

Since the proof for the gradient-based algorithm (3) follows trivially from the results of Bai et al. (2023); Wang et al. (2024b), we focus on metric-based algorithms (4) and (5), as well as amortization-based algorithms (6). We prove that there exists an TF model with specific real-valued parameters that can perform these algorithms.

Note that, for simplicity in proving the expressiveness of ICL with transformer, we focus on the algorithm framework: we leave feature-level transformations with neural networks alone as they can occur in both ICL model and conventional meta-learners, and enjoy the same universal approximation property (Hornik et al., 1989); we also do not consider the order of samples in $\mathcal{D}$, omitting any sequential models in meta-learners and positional embeddings in ICL. Typical metric-based algorithms are thus categorized into to types: one is based on pair-wise distance (4), e.g., MatchNet; another one is based on distance with class prototypes (5), e.g., ProtoNet. And typical amortization-based algorithms are summarized as a function taking the query and the encoded set as input (6). By proving that these exemplar set and inference functions can be implemented, more complex algorithms—such as feature-wise transformations, interactions between samples, and advanced distance functions—can be easily achieved through the recursive application of self-attention and feed-forward layers.

### C.1.3 ICL CAN PERFORM PAIR-WISE METRIC-BASED ALGORITHMS

For pair-wise metric-based algorithms, we take MatchNet for example, proving (12) can perform

$$\hat{\mathbf{y}}^{(n+1)} = \frac{1}{n}\sum_{i=1}^n <\mathbf{x}^{(i)}, \mathbf{x}^{(n+1)}> \mathbf{y}^{(i)}.\tag{17}$$

In fact, this case is relatively simple and can be implemented using a single-layer transformer without relying on the two tools. One implementation is $Q_0 = \begin{bmatrix} I & 0 \\ 0 & 0 \end{bmatrix}$, $P_0 = \begin{bmatrix} 0 & 0 \\ 0 & I \end{bmatrix}$, $W_r = \begin{bmatrix} 0 & \cdots & 0 \\ \boldsymbol{u}_1 & \cdots & \boldsymbol{u}_c \end{bmatrix}^\top$. Note that though the output of TF would be $\lambda \sum_{i=1}^n <\mathbf{x}^{(i)}, \mathbf{x}^{(n+1)}> \mathbf{y}^{(i)}$, where $\lambda \in \mathbb{R}$ is a query-specific value, it has the same classification result with (17).

### C.1.4 ICL CAN PERFORM CLASS-PROTOTYPE METRIC-BASED ALGORITHMS

For the second category of metric-based algorithms, we take ProtoNet for example, proving (12) can perform

$$\hat{\mathbf{y}}^{(n+1)} = \sum_{c=1}^C -||\frac{1}{N_c}\sum_{y^{(i)}=c}\mathbf{x}^{(i)} - \mathbf{x}^{(n+1)}||\boldsymbol{u}_c.\tag{18}$$

This can be implemented by a $3C - 1$ layer transformer achieving $[Z_{3l-1}]_{(d:d+2C),(n+1)} = \boldsymbol{q} + \sum_{c=1}^{l} ||\mathbf{x}^{(n+1)} - \boldsymbol{p}_c|| \boldsymbol{u}_{(c+1+C)mod(2C)}$ in the following step-by-step functions:

$$Z_0 = \begin{bmatrix} \mathbf{x}^{(1)} & \mathbf{x}^{(2)} & \cdots & \mathbf{x}^{(n)} & \mathbf{x}^{(n+1)} \\ \mathbf{y}^{(1)} & \mathbf{y}^{(2)} & \cdots & \mathbf{y}^{(n)} & \boldsymbol{q} \end{bmatrix}, \tag{19}$$

$$Z_1 = \begin{bmatrix} \mathbf{x}^{(i)} & \mathbf{x}^{(n+1)} - \boldsymbol{p}_1 \\ \mathbf{y}^{(i)} & \boldsymbol{q} \end{bmatrix}, \tag{20}$$

$$Z_2 = \begin{bmatrix} \mathbf{x}^{(i)} & \mathbf{x}^{(n+1)} - \boldsymbol{p}_1 \\ \mathbf{y}^{(i)} & \boldsymbol{q} + ||\mathbf{x}^{(n+1)} - \boldsymbol{p}_1|| \boldsymbol{u}_{C+2} \end{bmatrix}, \tag{21}$$

$$Z_3 = \begin{bmatrix} \mathbf{x}^{(i)} & \mathbf{x}^{(n+1)} \\ \mathbf{y}^{(i)} & \boldsymbol{q} + ||\mathbf{x}^{(n+1)} - \boldsymbol{p}_1|| \boldsymbol{u}_{C+2} \end{bmatrix}, \tag{22}$$

$$Z_4 = \begin{bmatrix} \mathbf{x}^{(i)} & \mathbf{x}^{(n+1)} - \boldsymbol{p}_2 \\ \mathbf{y}^{(i)} & \boldsymbol{q} + ||\mathbf{x}^{(n+1)} - \boldsymbol{p}_1|| \boldsymbol{u}_{C+2} \end{bmatrix}, \tag{23}$$

$$Z_5 = \begin{bmatrix} \mathbf{x}^{(i)} & \mathbf{x}^{(n+1)} - \boldsymbol{p}_2 \\ \mathbf{y}^{(i)} & \boldsymbol{q} + ||\mathbf{x}^{(n+1)} - \boldsymbol{p}_1|| \boldsymbol{u}_{C+2} + ||\mathbf{x}^{(n+1)} - \boldsymbol{p}_2|| \boldsymbol{u}_{C+3} \end{bmatrix}, \tag{24}$$

$$Z_6 = \begin{bmatrix} \mathbf{x}^{(i)} & \mathbf{x}^{(n+1)} \\ \mathbf{y}^{(i)} & \boldsymbol{q} + ||\mathbf{x}^{(n+1)} - \boldsymbol{p}_1|| \boldsymbol{u}_{C+2} + ||\mathbf{x}^{(n+1)} - \boldsymbol{p}_2|| \boldsymbol{u}_{C+3} \end{bmatrix}, \tag{25}$$

$$\cdots , \tag{26}$$

$$Z_{3l-3} = \begin{bmatrix} \mathbf{x}^{(i)} & \mathbf{x}^{(n+1)} \\ \mathbf{y}^{(i)} & \boldsymbol{q} + \sum_{c=1}^{l-1} ||\mathbf{x}^{(n+1)} - \boldsymbol{p}_i|| \boldsymbol{u}_{(i+1+C)mod(2C)} \end{bmatrix}, \tag{27}$$

$$Z_{3l-2} = \begin{bmatrix} \mathbf{x}^{(i)} & \mathbf{x}^{(n+1)} - \boldsymbol{p}_l \\ \mathbf{y}^{(i)} & \boldsymbol{q} + \sum_{c=1}^{l-1} ||\mathbf{x}^{(n+1)} - \boldsymbol{p}_i|| \boldsymbol{u}_{(i+1+C)mod(2C)} \end{bmatrix}, \tag{28}$$

$$Z_{3l-1} = \begin{bmatrix} \mathbf{x}^{(i)} & \mathbf{x}^{(n+1)} - \boldsymbol{p}_l \\ \mathbf{y}^{(i)} & \boldsymbol{q} + \sum_{c=1}^{l} ||\mathbf{x}^{(n+1)} - \boldsymbol{p}_i|| \boldsymbol{u}_{(i+1+C)mod(2C)} \end{bmatrix}, \tag{29}$$

$$\tag{30}$$

and readout by $W_r = \begin{bmatrix} 0 & \cdots & 0 & \cdots & 0 \\ -\boldsymbol{u}_{C+2} & \cdots & -\boldsymbol{u}_{(c+1+C)mod(2C)} & \cdots & -\boldsymbol{u}_1 \end{bmatrix}^{\top}$. Each step of function from $Z_l$ to $Z_{l+1}$ can be implemented by one transformer layer, which would be proved later.

### C.1.5 ICL Can Perform Amortization-Based Algorithms

Denote the set embedding $\frac{1}{n} \sum_{i=1}^{n} [\mathbf{x}^{(i)} | \mathbf{y}^{(i)}]$ as $\mathbf{e} \in \mathbb{R}^{d'}$. As $f$ in (6) can always be implemented by feed forward layers taking the concatenation of $\mathbf{x}^{(q)}$ and $\mathbf{e}$ as input, there exists a learnable function $h_1$ in $\mathbb{R}^{d'} \times \mathbb{R}^d$ and $h_2$ in $\mathbb{R}^d \times \mathbb{R}^{2C}$ that

$$f([(\mathbf{x}^{(q)})^{\top}, \mathbf{e}^{\top}]^{\top}) = h_2(\mathbf{x}^{(q)} + h_1(\mathbf{e})). \tag{31}$$

Thus, we prove that (12) can perform

$$\hat{\mathbf{y}}^{(n+1)} = h_2(\mathbf{x}^{(n+1)} + h_1(\frac{1}{n} \sum_{i=1}^{n} [\mathbf{x}^{(i)} | \mathbf{y}^{(i)}])). \tag{32}$$

This can be implemented by a 3 layer transformer achieving the following step-by-step function:

$$Z_0 = \begin{bmatrix} \mathbf{x}^{(1)} & \mathbf{x}^{(2)} & \cdots & \mathbf{x}^{(n)} & \mathbf{x}^{(n+1)} \\ \mathbf{y}^{(1)} & \mathbf{y}^{(2)} & \cdots & \mathbf{y}^{(n)} & \boldsymbol{q} \end{bmatrix}, \tag{33}$$

$$Z_1 = \begin{bmatrix} \mathbf{x}^{(i)} & \mathbf{x}^{(n+1)} + h_1(\frac{1}{n} \sum_{i=1}^{n} [\mathbf{x}^{(i)} | \mathbf{y}^{(i)}]) \\ \mathbf{y}^{(i)} & \boldsymbol{q} \end{bmatrix} \tag{34}$$

$$Z_2 = \begin{bmatrix} \mathbf{x}^{(i)} & h_2(\mathbf{x}^{(n+1)} + h_1(\frac{1}{n} \sum_{i=1}^{n} [\mathbf{x}^{(i)} | \mathbf{y}^{(i)}])) \\ \mathbf{y}^{(i)} & \boldsymbol{q} \end{bmatrix}, \tag{35}$$

and readout by $W_r = \begin{bmatrix} I & 0 \\ 0 & 0 \end{bmatrix}$. Each step of function from $Z_l$ to $Z_{l+1}$ can be implemented by one transformer layer, which would be proved now.

### C.1.6 THE FUNCTION OF ONE TRANSFORMER LAYER

A transformer layer (11) can perform a wide range of functions, as we can decompose it is composed of a self-attention layer and feed-forward layers, where (i) self-attention (16) with orthonormal label tokenization (15) can achieve a wide range of label-aware set operations. (ii) feed-forward layer $\sigma(\cdot)$ in (11) can learn any measurable functions in $\mathbb{R}^{d+2c} \times \mathbb{R}^{d+2c}$. Here we prove how a transformer layer can obtain the above functions from $Z_l$ to $Z_{l+1}$. The main idea is a function can be decomposed to three sub-steps: label-selecting which is achieved by $A = Z^\top Q Z$, linear interaction achieved by $PZ\mathrm{norm}_1^{\mathrm{col}}(A)$, and non-linear transformation by $\sigma_\theta(\cdot)$ if needed.

**Label-Aware Attention.** In one self-attention layer, equation (16), first each column in $Z$ refer to other columns through attention weights $A = Z^\top Q Z$. $A \in \mathbb{R}^{(n+1)\times(n+1)}$ is selecting interaction objectives and weighting interaction weights. We use *label-aware* to describe $\{A \in \mathbb{R}^{(n+1)\times(n+1)} \mid (\mathbf{y}^{(i)} = \mathbf{y}^{(i')}) \wedge (\mathbf{y}^{(j)} = \mathbf{y}^{(j')}) \Rightarrow A_{i,j} = A_{i',j'}\}$, i.e., the interaction weight between ordered sample-pair $(i, j)$ only depends on their labels $(\mathbf{y}^{(i)}, \mathbf{y}^{(j)})$ (including unknown label $\boldsymbol{q}$), and can be arbitrary value in $\mathbb{R}$.

With our orthonormal label embedding in $\mathbb{R}^{2c}$, $A$ is label-aware, thus can achieve label-aware interaction. For example, to achieve (19) to (20), we require $(c+1)^2$ conditions about $A$:

$$\begin{cases} A_{c_1,q} = 1 \\ A_{c_i,q} = 0, \ i \in \{2, 3, \cdots, C\} \\ A_{q,c_i} = 0, \ i \in \{1, 2, \cdots, C\} \\ A_{c_i,c_j} = 0, \ i,j \in \{1, 2, \cdots, C\} \\ A_{q,q} = 0 \end{cases} \tag{36}$$

As $A = Z^\top Q Z$ and $A_{i,j}$ is only related to $\mathbf{y}^{(i)}, \mathbf{y}^{(j)}$, we have $Q = \begin{bmatrix} 0 & 0 \\ 0 & L \end{bmatrix}$ where $L \in \mathbb{R}^{2C \times 2C}$. Equation (36) gives $(C+1)^2$ linear equations about $L$:

$$\begin{cases} \boldsymbol{u}_1^\top L \boldsymbol{u}_q = 1 \\ \boldsymbol{u}_i^\top L \boldsymbol{u}_q = 0, \ i \in \{2, 3, \cdots, C\} \\ \boldsymbol{u}_q^\top L \boldsymbol{u}_i = 0, \ i \in \{1, 2, \cdots, C\} \\ \boldsymbol{u}_i^\top L \boldsymbol{u}_j = 0, \ i,j \in \{1, 2, \cdots, C\} \\ \boldsymbol{u}_q^\top L \boldsymbol{u}_q = 0 \end{cases} \tag{37}$$

**Proposition C.2.** $\forall c > 1$, *Equation (37) has solutions in* $\mathbb{R}^{2C \times 2C}$.

*Proof.* Denote $\boldsymbol{u}_i \otimes \boldsymbol{u}_j = [u_{i,1}\boldsymbol{u}_j^\top, u_{i,2}\boldsymbol{u}_j^\top, \cdots, u_{i,2C}\boldsymbol{u}_j^\top]^\top \in \mathbb{R}^{4C^2}$, $\vec{L} = [L_{1,1}, \cdots, L_{1,2C}, \cdots, L_{2C,2C}]^\top \in \mathbb{R}^{4C^2}$, then $\boldsymbol{u}_i^\top L \boldsymbol{u}_j = (\boldsymbol{u}_i \otimes \boldsymbol{u}_j)^\top \vec{L}$.

(37)$\Longleftrightarrow U\vec{L} = \vec{A}$, where

$$U = [\boldsymbol{u}_i \otimes \boldsymbol{u}_j \text{ for } i,j \in \{1, 2, \cdots, C+1\}]^\top \in \mathbb{R}^{(C+1)^2 \times 4C^2}. \tag{38}$$

$$(\boldsymbol{u}_i \otimes \boldsymbol{u}_j)^\top (\boldsymbol{u}_{i'} \otimes \boldsymbol{u}_{j'}) = \sum_{t=1}^{2c} u_{it} u_{i't} \boldsymbol{u}_j^\top \boldsymbol{u}_{j'} \tag{39}$$

$$= (\boldsymbol{u}_j^\top \boldsymbol{u}_{j'})(\sum_{t=1}^{2c} u_{it} u_{i't}) \tag{40}$$

$$= (\boldsymbol{u}_j^\top \boldsymbol{u}_{j'})(\boldsymbol{u}_i^\top \boldsymbol{u}_{i'}) \tag{41}$$

$$= \begin{cases} 1, \text{ if } i = i' \wedge j = j' \\ 0, \text{ if } i \neq i' \vee j \neq j' \end{cases} \tag{42}$$

$$\Longrightarrow \text{rank}(U) = (C+1)^2 \tag{43}$$

$$[U, \vec{A}] \in \mathbb{R}^{(C+1)^2 \times (4C^2+1)}, c > 1 \Longrightarrow \text{rank}([U, \vec{A}]) \le (C+1)^2. \tag{44}$$

$(43), (44) \Longrightarrow$

$$\text{rank}([U, \vec{A}]) = \text{rank}(U) = (C+1)^2 < 4C^2. \tag{45}$$

$\Longrightarrow$ Equation $U\vec{L} = \vec{A}$ has solutions in $R^{4C^2}$. $\Longleftrightarrow$ Equation (37) has solutions in $\mathbb{R}^{2C \times 2C}$.  $\square$

Note that for any function from $Z_l$ to $Z_{l+1}$, the number of conditions about $A \le (2C)^2$. Thus for any label-aware function from $Z_l$ to $Z_{l+1}$, it requires a label-ware $A$ and we can find a linear system of equations $U\vec{L} = \vec{A}$, that has solutions in $\mathbb{R}^{4c^2}$, as the proof $\text{rank}([U, \vec{A}]) = \text{rank}(U) \le 4c^2$ is without loss of generalizability.

**Linear Interaction.** After obtaining desired $A$, $\text{norm}_1^{\text{col}}(A)$ is performed as $\text{norm}_1^{\text{col}}$ is a better approximation of $\text{softmax}$ than $\frac{1}{n}$, and also required to deal with inconsistent label number in our classification tasks. Then all columns in $Z$ interact with the others linearly through $PZ\text{norm}_1^{\text{col}}(A)$. Still taking (19) to (20) as example, after obtaining desired $A$ satisfying (36), $P = \begin{bmatrix} -I & 0 \\ 0 & 0 \end{bmatrix}$ can achieve the function.

**Non-Linear Transformation.** In (11), $\sigma_\theta(\cdot)$ is feed-forward layers that function on each column of $Z$ independently. Thanks to the universal approximation property (Hornik et al., 1989), it can approximate any measurable function in $\mathbb{R}^{d+2c} \times \mathbb{R}^{d+2c}$ to any desired degree of accuracy. Thus, feature-level non-linear transformation from $Z_l$ to $Z_{l+1}$ could turn to $\sigma_{\theta_l}(\cdot)$. For example, (19) to (20) does not require non-linearity so it can be implemented as $\sigma(\boldsymbol{z}) = \boldsymbol{z}$. For (20) to (21), one implementation is $\sigma(\boldsymbol{z}) = [0, ||[\boldsymbol{z}]_{1:d+1}||\boldsymbol{u}_{C+2}]^\top$. Note that in this step, the $=$ does not hold strictly, but can be approximated by MLPs with error $\epsilon > 0$. We use "$=$" to mean such approximation for simplicity, as the error can be arbitrary small.

In conclusion, each step from $Z_l$ to $Z_{l+1}$ can be implemented using a transformer layer. Typical metric- and amortization-based meta-learning algorithms (4)(5)(6) can be implemented with ICL. More complex models following the same set functions can also be performed by ICl with additional recursion of transformer layers, whose proof is trivial. Moreover, as it has been proved that ICL can perform gradient-based algorithms (3), ICL can exactly perform conventional handcrafted meta-learning algorithms.

## D  GENERATING TASKS

Here, we present the task generation process for $g_{\text{sim}}$, $g_{\text{prt}}$ and $g_{\text{am}}$. The tasks are designed in specific forms such that they are all linearly separable in $\mathbf{x}^{(i)} \in \mathbb{R}^d$, enabling 2D visualization to observe the behavior of ICL.

For pair-wise metric-based algorithms $g_{\text{sim}}$, we generate a task by sampling $C \times N_C$ support samples $\{\mathbf{x}^{(i)} \in \mathbb{R}^d\}_{i=1}^{C \times N_C}$ from distribution $p(\tau)$ and randomly assign them with labels $y^{(i)} = c$, making $C \times N_C$ supports exactly contains $N_C$ label $c$ for each $c = 1, 2, \cdots, C$. Then the remaining samples $\{\mathbf{x}^{(i)} \in \mathbb{R}^d\}_{i=C \times N_C+1}^{N_C+N}$ are sampled from distribution $p_x$, and assigned with labels $y^{(i)} = \arg\max_c \sum_{j=1, y^{(j)}=c}^{C \times N_C} < \mathbf{x}^{(i)}, \mathbf{x}^{(j)} >$. A typical meta-learner, MatchNet, can learn the optimal classifier. A case is shown in Figure 9(a), where each point corresponds to a $\mathbf{x}^{(i)} \in \mathbb{R}^2$ and different labels are assigned with different colors.

For class-prototype metric-based algorithms $g_{\text{prt}}$, we generate a task by sampling $C$ prototypes $\{\boldsymbol{p}_c \in \mathbb{R}^d\}_{c=1}^{C}$ from $p(\tau)$. Then sample $\{\mathbf{x}^{(i)} \in \mathbb{R}^d\}_{i=1}^{N}$ from $p_x$, and assign labels by $y^{(i)} = \arg\min_c ||\boldsymbol{p}_c - \mathbf{x}^{(i)}||$. The corresponding optimal classifier is ProtoNet. A case is shown in Figure 9(b).

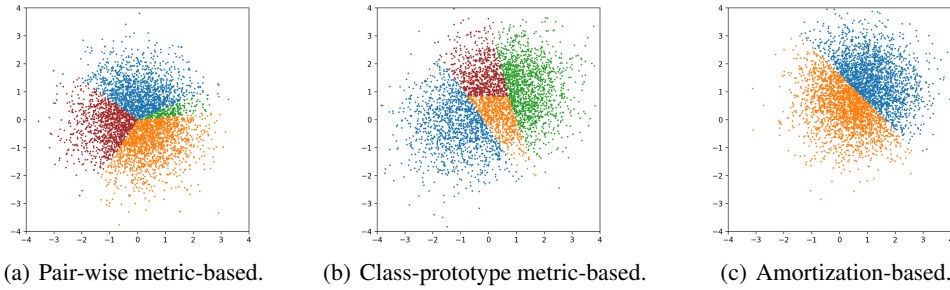

(a) Pair-wise metric-based.    (b) Class-prototype metric-based.    (c) Amortization-based.

Figure 9: Examples of three types of tasks.

For amortization-based algorithms $g_{am}$, we pre-define a partition of $R$, $\{\Omega_c\}_{c=1}^{C}$, as decision range. We generate a task by sampling $\boldsymbol{\mu} \in \mathbb{R}^d$ from $p(\tau)$. Then sample $\{\mathbf{x}^{(i)} \in \mathbb{R}^d\}_{i=1}^{N}$ from $p_x(\boldsymbol{\mu}) = \mathcal{N}(\boldsymbol{\mu}, \Sigma)$, and assign labels by $y^{(i)} = c$ where $\sum_{t=1}^{d}[\mathbf{x}^{(i)} - \boldsymbol{\mu}]_t \in \Omega_c$. The corresponding optimal classifier is CNPs. A case is shown in Figure 9(c).

# E    MORE EMPIRICAL RESULTS

## E.1    ICL MODEL TRAINED WITH SINGLE TYPE OF TASKS

Figures 10, 11 and 12 show more cases to support that ICL model learns MatchNet, ProtoNet, CNPs on $\boldsymbol{\Omega}_{sim}$, $\boldsymbol{\Omega}_{prt}$, $\boldsymbol{\Omega}_{am}$ respectively.

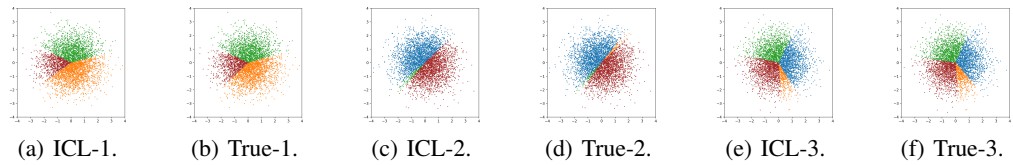

(a) ICL-1.    (b) True-1.    (c) ICL-2.    (d) True-2.    (e) ICL-3.    (f) True-3.

Figure 10: Comparison of ICL's predictions with true labels on pair-wise metric-based meta-testing tasks, with the ICL model trained on a single task type.

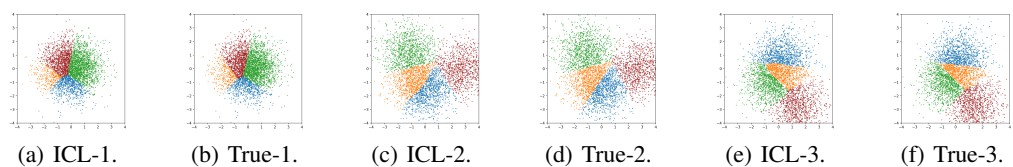

(a) ICL-1.    (b) True-1.    (c) ICL-2.    (d) True-2.    (e) ICL-3.    (f) True-3.

Figure 11: Comparing ICL's predictions and true labels on class-prototype metric-based tasks, with the ICL model trained on a single task type.

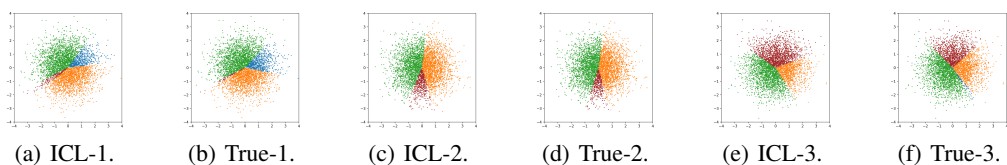

(a) ICL-1.    (b) True-1.    (c) ICL-2.    (d) True-2.    (e) ICL-3.    (f) True-3.

Figure 13: Comparing ICL's predictions and true labels on pair-wise metric-based tasks, with the ICL model trained on mixed task types.

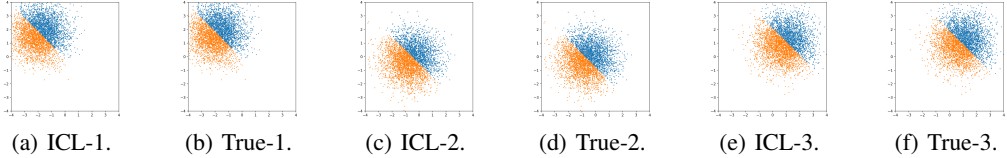

(a) ICL-1.    (b) True-1.    (c) ICL-2.    (d) True-2.    (e) ICL-3.    (f) True-3.

Figure 12: Comparing ICL's predictions and true labels on amortization-based tasks, with the ICL model trained on a single task type.

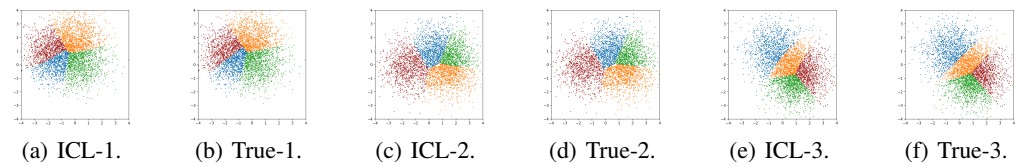

(a) ICL-1.    (b) True-1.    (c) ICL-2.    (d) True-2.    (e) ICL-3.    (f) True-3.

Figure 14: Comparing ICL's predictions and true labels on class-prototype metric-based tasks, with the ICL model trained on mixed task types.

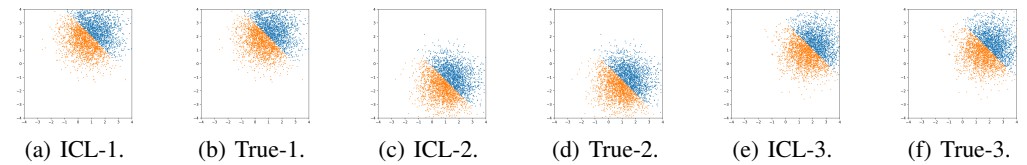

(a) ICL-1.    (b) True-1.    (c) ICL-2.    (d) True-2.    (e) ICL-3.    (f) True-3.

Figure 15: Comparing ICL's predictions and true labels on amortization-based tasks, with the ICL model trained on mixed task types.

### E.2 ICL MODEL TRAINED WITH MIXED TYPE OF TASKS

Figures 13, 14 and 15 show cases to support that ICL model learns data-dependent optimal learning algorithm on $\Omega_{\mathrm{mix}} = \{\Omega_{\mathrm{sim}}, \Omega_{\mathrm{prt}}, \Omega_{\mathrm{am}}\}$.

## F  EXPERIMENT DETAILS

Our code is provided at `https://github.com/ovo67/Uni_ICL`.

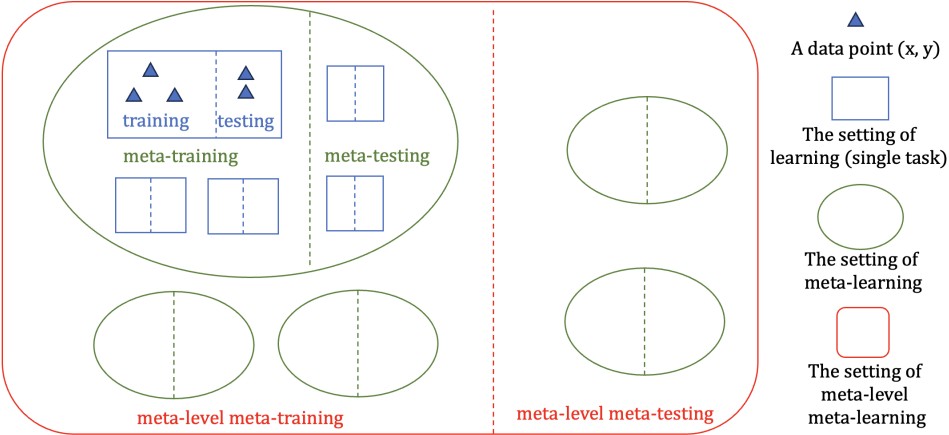

Figure 16: The problem setting of meta-level meta-learning.

## G   ILLUSTRATION OF META-LEVEL META-LEARNING

Here, we provide further illustrations of the problem setting and the algorithm procedure of the proposed meta-level meta-learning, supplementing Section 4.1. The problem setting is illustrated in Figure 16. Conventional single-task learning solves problems on a single task, which consists of a set of data points for training and another set for testing. Typical meta-learning addresses problems on a domain, which includes a set of tasks for meta-training and another set for meta-testing. Meta-level meta-learning operates on a collection of domains, each containing multiple tasks, and involves a set of domains for meta-level meta-training and another set for meta-level meta-testing.

Next, we illustrate the process of training an $M^2$-ICL model. The meta-level meta-training process, with MAML as the meta-level meta-learner, is provided in Algorithm 1. Note that the meta-level meta-learner $G(;\Delta)$ in Equation (8) does not have to be MAML. We take MAML as $G(;\Delta)$ because of its model-agnostic property, which allows us to avoid designing additional learnable parameters $\Delta$. However, one can design customized $G(;\Delta)$ meta-learners, such as metric-based or amortization-based methods.

---

**Algorithm 1** Training $M^2$-ICL

**Input:** Training domain distribution $p(\delta)$, ICL model $g(\cdot;\theta)$.
 1: **while** Not converge **do**
 2:     Sample a domain $\delta \sim p(\delta)$.
 3:     Sample tasks $\tau \sim p_\delta(\delta)$ to form task sets $\mathbf{\Omega}_\delta^{\text{tr}}$ and $\mathbf{\Omega}_\delta^{\text{val}}$.
 4:     **for** Every task $\tau \in \mathbf{\Omega}_\delta^{\text{tr}}$ **do**
 5:         Calculate task loss $\ell_{\text{meta}}(\tau, g(\cdot;\theta)) = \frac{1}{N_\tau}\sum_{i=0}^{N_\tau-1} \ell(\hat{\mathbf{y}}^{(i+1)}, \mathbf{y}^{(i+1)})$ by (12).
 6:     **end for**
 7:     Update $\theta_\delta = \theta - \nabla_\theta \frac{1}{|\mathbf{\Omega}_\delta^{\text{tr}}|}\sum_{\tau \in \mathbf{\Omega}_\delta^{\text{tr}}} \ell_{\text{meta}}(\tau, g(\cdot;\theta))$.
 8:     **for** Every task $\tau \in \mathbf{\Omega}_\delta^{\text{val}}$ **do**
 9:         Calculate task loss $\ell_{\text{meta}}(\tau, g(\cdot;\theta_\tau)) = \frac{1}{N_\tau}\sum_{i=0}^{N_\tau-1} \ell(\hat{\mathbf{y}}^{(i+1)}, \mathbf{y}^{(i+1)})$ by (12).
10:     **end for**
11:     Update $\theta \leftarrow \theta - \nabla_\theta \frac{1}{|\mathbf{\Omega}_\delta^{\text{val}}|}\sum_{\tau \in \mathbf{\Omega}_\delta^{\text{val}}} \ell_{\text{meta}}(\tau, g(\cdot;\theta_\tau))$.
12: **end while**

---

## H   META-LEVEL META-LEARNING FOR CROSS-DOMAIN FEW-SHOT IMAGE CLASSIFICATION

We investigate the effective of meta-level meta-learning on real-world few-shot image classification problem. We use META-DATASET (Triantafillou et al., 2020) for training. Because it contains

multiple datasets inside each we can sample many few-shot classification tasks, thus can be naturally divided into multiple domains to perform the meta-level meta-training (8).

Following standard settings, we used the training sets of ILSVRC, Omniglot, Aircraft, Birds, Textures, Quick Draw, and Fungi during training. For testing, we used unseen datasets such as Traffic Signs, MSCOCO, and additional datasets like MNIST, CIFAR10, and CIFAR100. Each dataset is treated as a domain for meta-level meta-learning. We considered 5-way 5-shot tasks at the meta-level and 8/16/32 tasks-for-adaptation per domain at the meta-meta-level. The sampling of classes and images to form tasks, as well as the sampling of tasks-for-adaptation within a domain, was random.

We consider the following baselines:

- ICL w/o adpt: The standard meta-learning setting, where meta-training is performed on all tasks without distinguishing between datasets. During meta-testing, no domain adaptation is performed, meaning that the 8/16/32 tasks are not utilized.

- ICL w/ adpt: The meta-training process is identical to that of ICL w/o adpt. While during meta-testing, the model adapts using 8/16/32 domain-specific tasks by fine-tuning all parameters (step = 5, learning rate = 0.0001, batch size = 8/16/32).

- $M^2$-ICL: Meta-level meta-training an ICL model following the method introduced in Section 4.1. The domain adaptation process, i.e., the inner-loop of meta-level-MAML is configured as step=5, lr=0.0001, with 16 tasks-for-adaptation per domain. During testing, given 8/16/32 domain-specific tasks, the same adaptation process is applied.

Our implementation builds the ICL model with a 8-layer transformer (without positional encoding), where the input features are 512-dim extracted by a ResNet (i.e., resnet-18). Though the model is relatively toy, it is enough to verify the effectiveness of meta-level meta-learning for improving ICL in real-world application: the method pipeline is generalizable and one can replace them with models with more advanced architectures or for other applications.

Table 1: Image classification accuracy (%) on META-DATASET, a benchmark for cross-domain few-shot image classification.

| Method | Traffic Signs | MSCOCO | MNIST | CIFAR10 | CIFAR100 | Average |
|---|---|---|---|---|---|---|
| ICL w/o adpt | 45.4 | 35.5 | 88.1 | 65.2 | 55.9 | 58.02 |
| ICL w/ adpt (8 tasks) | 41.9 | 35.1 | 76.4 | 64.9 | 55.5 | 53.76 |
| ICL w/ adpt (16 tasks) | 43.3 | 36.2 | 78.5 | 66.0 | 56.3 | 56.06 |
| ICL w/ adpt (32 tasks) | 46.1 | 36.5 | 83.2 | 66.8 | 58.3 | 58.18 |
| $M^2$-ICL (8 tasks) | 45.9 | 39.4 | 86.6 | 67.4 | 57.2 | 59.30 |
| $M^2$-ICL (16 tasks) | 47.5 | 40.6 | 88.9 | 68.0 | 57.9 | 60.58 |
| $M^2$-ICL (32 tasks) | 52.6 | 44.1 | 91.0 | 69.4 | 59.2 | **63.26** |

The results are provided in Table 1. We find that the $M^2$-ICL significantly outperforms ICL w/o or w/ adpt with any tasks. Specifically, comparing with ICL w/o adpt (standard meat-training and testing), adapting the ICL model with 8 tasks badly harms the performance due to overfitting, and with 16 tasks also do harm, while 32 tasks shows marginally improvement. However, adapting the $M^2$-ICL model with only 8 tasks is enough to surpasses the average performance, and the growing number of tasks for adaptation brings more significant improvement. ICL w/o adpt, ICL w/ adpt (32 tasks) and $M^2$-ICL (8 tasks) have comparable performance. This show the proposed meta-level meta-learning is very effective to improve the few-task domain adaptation ability.

