# OpenReview forum: "Why In-Context Learning Models are Good Few-Shot Learners?"
_ICLR.cc/2025/Conference — ICLR 2025 Poster_

### Official Review · Reviewer_bz7Z · 2024-10-24

**Soundness:** 1
**Presentation:** 1
**Contribution:** 1
**Rating:** 3
**Confidence:** 4

**Summary:**

This paper investigates the meta-learning capabilities of in-context learners (ICLs) by comparing them to state-of-the-art meta-learning algorithms. The authors demonstrate how ICL can learn optimal algorithms in various scenarios and propose a strategy to enhance generalization by incorporating techniques from supervised learning. They also explore the impact of applying curriculum learning during the training phase.

**Strengths:**

*  The observation that different meta-learning methods generalize best on their respective task sets is interesting. The paper shows that ICL, when trained on a mixture of task sets, can outperform individual meta-learners.
* The study of generalization across tasks is valuable for understanding the capabilities of ICL in various domains.

**Weaknesses:**

*  The paper is poorly written, with numerous typos and a lack of clarity, making it difficult to follow. A complete revision is necessary.
*  The related work section needs significant improvement. Rather than simply listing various meta-learning studies, it should focus on situating this work within the broader literature, especially by incorporating more relevant references on in-context learning and model generalization.
*  Several key references are missing in the text, including citations for MatchNet, ProtoNet and CNP in line 244.
*  There are incorrect assumptions present throughout the paper. For example, lines 82–85 incorrectly claim that transformers are inherently permutation-invariant, and line 453 mistakenly implies that all in-context learners are pre-trained, whereas studies such as [1] and [2] demonstrate otherwise.
* The main text and the appendix feel disconnected, almost as if they belong to separate studies. Mathematical proofs, even if valuable, bring a little contribution if left in the appendix.
*  The experimental design is weak, and there is no reference to standard literature or pipelines for task creation. The authors should either adopt state-of-the-art experimental methodologies or provide a strong justification for deviating from established practices.
*  More details about the task generation process  should be included in the main text, not only in the appendix section. This would give readers a better understanding of the experimental setup and strengthen the paper’s presentation.
*  The results presented in Figure 7, which show that adapting a model to a different distribution leads to higher squared error, seem trivial. Furthermore, the meta-meta-learner does not outperform in a meaningful way, so the contribution of this section is unclear.
*  The experiments with curriculum learning are insufficient to demonstrate its effectiveness in improving in-context learning. A more robust and comprehensive evaluation is needed to validate these claims.
*  What is the meaning of Figure 1? What guarantees the orthogonality among the three methods?
*  In the first part of the paper, the authors claim that ICL can act as different types of meta-learners (gradient/metric/amortization-based) and learn the "optimal" algorithm. However, the notion of "optimal" is vague and needs to be better defined within the context of the study.



References: \
[1] Sewon Min, Mike Lewis, Luke Zettlemoyer, & Hannaneh Hajishirzi. (2022). MetaICL: Learning to Learn In Context.\
[2] Kirsch, Louis, et al. "General-purpose in-context learning by meta-learning transformers." arXiv preprint arXiv:2212.04458 (2022).

**Questions:**

See Weaknesses part.

---

> ### Author Response · Authors · 2024-11-20
> **Part I**
>
> Thank you for your comments and suggestions. We appreciate the opportunity to clarify and address potential misunderstandings in this discussion.
>
> ## 1. Numerous typos and a lack of clarity; add citations for MatchNet, ProtoNet and CNP in line 244.
> We sincerely apologize for the typos that may have hindered your ability to fully engage with the paper. We greatly appreciate your careful reading and constructive feedback. In the revised manuscript, we have corrected the typos and added the citations for MatchNet, ProtoNet, and CNP on line 244.
>
>
> ## 2. lines 82–85 incorrectly claim that transformers are inherently permutation-invariant
>
> This appears to be a misunderstanding.
>
> First, in lines 82–85, the term “permutation-invariant” specifically describes “each layer,” referring to the combination of a self-attention module and a feedforward layer, which together form the basic building block of a transformer. Both of these modules process input as a set of vectors and are inherently permutation-invariant.
>
> Second, a transformer becomes permutation-variant only when positional embeddings are added to the input, which introduces order sensitivity. However, using positional embeddings is not feasible when investigating a learning algorithm, as it would enforce a sequential structure on the training set. In this paper, we focus on using transformers as learning algorithms, where the natural training set does not exhibit sequential structure. Therefore, as mentioned in Appendix B and discussed in Section 6, positional embeddings are omitted. This approach aligns with the common practice in existing works aiming to understand ICL, such as [1][2][3].
> [1] Transformers Learn In-Context by Gradient Descent, ICML 2023
>
> [2] Transformers learn to implement preconditioned gradient descent for in-context learning, NeurIPS 2023
>
> [3] Can Looped Transformers Learn to Implement Multi-step Gradient Descent for In-context Learning?, ICML 2024
>
>
> ## 3.line 453 mistakenly implies that all in-context learners are pre-trained, whereas studies such as [1] and [2] demonstrate otherwise".
>
> This appears to be a misunderstanding.
>
> Both [1] and [2] follow Eqn (13) to pre-train their ICL models. The key difference is that [1] and [2] restrict the pre-training prompts to consist of only a few examples per task, whereas typical LLM pre-training does not strictly adhere to this structure. Nonetheless, all these methods involve pre-training under the global objective described in Eqn (13), which can also be interpreted as a form of meta-training.
>
> In contrast, our work introduces an additional layer of meta-training at the domain level, termed meta-level meta-learning. The relationship between ICL and  M^2 -ICL is analogous to the relationship between a model trained with standard supervised learning and a model meta-trained using MAML.
>
> [1] Sewon Min, Mike Lewis, Luke Zettlemoyer, & Hannaneh Hajishirzi. (2022). MetaICL: Learning to Learn In Context.
>
> [2] Kirsch, Louis, et al. "General-purpose in-context learning by meta-learning transformers." arXiv preprint arXiv:2212.04458 (2022).
>
> ## 4. The related work section needs significant improvement. Rather than simply listing various meta-learning studies, it should focus on situating this work within the broader literature, especially by incorporating more relevant references on in-context learning and model generalization.
> In our submitted manuscript, we had addressed existing works on various perspectives of understanding ICL in the Introduction. We had discussed a broader range of related works in Appendix A, including topics such as ICL, meta-learning, the expressiveness of neural networks, and the generalizability of deep learning. In these sections, we have explicitly outlined the connections between these works and the contributions of this paper.

---

> > ### Author Response · Authors · 2024-11-20
> > **Part II**
> >
> > ## 5. The results presented in Figure 7, which show that adapting a model to a different distribution leads to higher squared error, seem trivial. Furthermore, the meta-meta-learner does not outperform in a meaningful way, so the contribution of this section is unclear.
> > The effectiveness of the proposed meta-meta-learner is demonstrated in Figure 7, particularly through the consistent advantage of  M^2 -ICL w/ adpt over both ICL w/ adpt and ICL w/o adpt in each sub-figure. This advantage is most pronounced when the number of adaptation tasks is very small, as shown in Figure 7(a).
> >
> > As introduced in Section 5.1, a widely discussed challenge in leveraging LLMs is how to adapt them to specific domains with limited data, combining general intelligence with domain-specific knowledge. This few-data adaptation problem often leads to overfitting. In conventional supervised learning, such problems can be addressed through meta-learning. Given that an ICL model is inherently a meta-learner, and its pretraining process can be viewed as meta-training, we extend this by adding an additional layer of meta-training at the domain level, leveraging the fact that large-scale pretraining data can be divided into multiple semantically defined domains. This approach, termed meta-level meta-learning, parallels the relationship between standard supervised learning and MAML in meta-learning.
> >
> > Figure 7 compares different solutions under the scenario of pretraining on a large-scale dataset and testing on an out-of-distribution domain with a limited number of tasks (64, 256, 1024) available for adaptation. The considered solutions include:
> >
> > - **ICL w/ adpt**: Performs well with sufficient data but is prone to overfitting when the data is limited.
> > - **ICL w/o adpt**: Avoids overfitting but struggles with a lack of adaptation.
> > - **M^2 -ICL w/ adpt**: Addresses the overfitting problem by modifying the pretraining process.
> >
> > Across all sub-figures in Figure 7,  M^2 -ICL w/ adpt consistently outperforms the other methods, particularly when the number of adaptation tasks is very small (Figure 7(a)).
> >
> > We have also made minor revisions to the explanation of these results in the revised manuscript to enhance clarity and understanding. The revised sentences in pdf are highlighted in blue.
> >
> > ## 6. The experimental design is weak, and there is no reference to standard literature or pipelines for task creation. The authors should either adopt state-of-the-art experimental methodologies or provide a strong justification for deviating from established practices. The experiments with curriculum learning are insufficient to demonstrate its effectiveness in improving in-context learning. A more robust and comprehensive evaluation is needed to validate these claims.
> >
> > Regarding the experiments in Section 4, our design aims to support novel comparisons between ICL and typical meta-learners, which have not been extensively explored in the existing literature on ICL or meta-learning. As such, explicit references to standard methodologies are not applicable here.
> >
> > For the experiments in Section 5, our goal is not to achieve state-of-the-art (SOTA) performance on established benchmarks, as we are not proposing a new end-to-end method or model. Instead, our focus is on demonstrating the feasibility of transferring deep-learning techniques from supervised learning to meta-learning to enhance ICL. To this end, we showcase the effectiveness of our proposed strategies, including meta-level meta-learning and meta-level curriculum learning, through targeted examples.
> >
> > In response to this comment, we have added additional experiments to demonstrate the effectiveness of our methods and discuss their applicability to real-world datasets. These new results further validate our claims and provide stronger evidence for the practical relevance of the proposed strategies.
> >
> > **Meta-level meta-learning:**
> >
> > We have evaluated the effectiveness of meta-level meta-learning on a real-world few-shot image classification problem using the Meta-Dataset [1]. This dataset is particularly suitable as it contains multiple sub-datasets, each of which can be used to sample many few-shot classification tasks, making it naturally divisible into multiple domains for meta-level meta-training (Eqn (8)).
> >
> > Following standard settings, we used the training sets of ILSVRC, Omniglot, Aircraft, Birds, Textures, Quick Draw, and Fungi during training. For testing, we used unseen datasets such as Traffic Signs, MSCOCO, and additional datasets like MNIST, CIFAR10, and CIFAR100. Each dataset is treated as a domain for meta-level meta-learning.
> > We considered 5-way 5-shot tasks at the meta-level and 8/16/32 tasks-for-adaptation per domain at the meta-meta-level. The sampling of classes and images to form tasks, as well as the sampling of tasks-for-adaptation within a domain, was random.

---

> ### Author Response · Authors · 2024-11-20
> **Part III**
>
> We consider the following baselines:
> *  **ICL w/o adpt**: The standard meta-learning setting, where meta-training is performed on all tasks without distinguishing between datasets. During meta-testing, no domain adaptation is performed, meaning that the 8/16/32 tasks are not utilized.
> *  **ICL w/ adpt**: The meta-training process is identical to that of ICL w/o adpt. While during meta-testing, the model adapts using 8/16/32 domain-specific tasks by fine-tuning all parameters (step = 5, learning rate = 0.0001, batch size = 8/16/32).
> *  **M $^2$-ICL**: Meta-level meta-training of an ICL model using the method introduced in Section 5.1. The domain adaptation process (i.e., the inner loop of meta-level MAML) is configured with step = 5, learning rate = 0.0001, and 16 tasks-for-adaptation per domain. Meta-meta-training and evaluation were conducted for 8/16/32 tasks-for-adaptation, respectively. During testing, the same adaptation process was applied.
>
> Our implementation utilized an 8-layer transformer (without positional encoding) as the ICL model, with input features extracted by a ResNet-18 (512 dimensions). Although this model is relatively simple, it is sufficient to verify the effectiveness of meta-level meta-learning in improving ICL for real-world applications. The pipeline is generalizable, and the architecture can be replaced with more advanced models or adapted for other applications.
>
> The results, presented in the table below and inlcuded in Appendix H of the revised manuscript, demonstrate that M $^2$-ICL significantly outperforms both ICL w/o adpt and ICL w/ adpt across all task settings. Specifically, adapting the ICL model with only 8 tasks often results in overfitting, while using 16 tasks shows limited improvement, and 32 tasks leads to marginal gains. In contrast, adapting M $^2$-ICL with just 8 tasks surpasses the average performance of other methods, and increasing the number of tasks for adaptation brings further improvements. Notably, ICL w/o adpt, ICL w/ adpt (32 tasks), and M $^2$-ICL (8 tasks) show comparable performance. These results highlight the effectiveness of the proposed meta-level meta-learning approach in enhancing few-task domain adaptation.
>
> Method | Traffic Signs| MSCOCO | MNIST | CIFAR10| CIFAR100 | Average
> ------------- | -------------|--|--|--|--|--|
> ICL w/o adpt  | 45.4 |35.5|88.1 |65.2|55.9 |58.02
> ICL w/ adpt (8 tasks) | 41.9 |35.1|76.4 |64.9|55.5|53.76
> ICL w/ adpt (16 tasks) | 43.3|36.2|78.5|66.0|56.3|56.06
> ICL w/ adpt (32 tasks) | 46.1|36.5|83.2|66.8|58.3|58.18
> M $^2$-ICL (8 tasks)| 45.9|39.4|86.6|67.4|57.2|59.30
> M $^2$-ICL (16 tasks)| 47.5|40.6|88.9|68.0|57.9|60.58
> M $^2$-ICL (32 tasks)| **52.6**|**44.1**|**91.0**|**69.4**|**59.2**|**63.26**
>
>
> **Meta-level curriculum learning:**
>
>
> For meta-level curriculum learning, there are existing works that demonstrate practical approaches with typical meta-learners on real-world problems. There are various strategies to determine which tasks should be learned in a given episode during meta-training. A common approach is knowledge-based curriculum learning, where task difficulty is defined using expert knowledge specific to the problem. For instance, [2] defines task difficulty based on interactions between semantically similar relations observed in a task sequence for continual relation extraction problems.
> Another approach is model-based curriculum learning, where task difficulty is determined based on the evaluation performance of the current model, as explored in [3][4]. More generally, to schedule task-sampling strategies in meta-learning, additional factors such as task diversity within a batch or other task-specific information can be considered, as proposed in [5][6][7].
> These strategies can be extended to the pretraining of ICL models for real-world problems by replacing the meta-learner with the ICL model. We believe adopting such strategies could help construct an effective curriculum for complex, real-world datasets.
>
> [1] Meta-Dataset: A dataset of datasets for learning to learn from few examples, ICLR 2020
>
> [2] Curriculum-meta learning for order-robust continual relation extraction, AAAI 2021
>
> [3] Curriculum meta-learning for next POI recommendation, KDD 21
>
> [4] Progressive Meta-Learning With Curriculum, 2022
>
> [5] Meta-Curriculum Learning for Domain Adaptation in Neural Machine Translation, AAAI 2021
>
> [6] Adaptive Task Sampling for Meta-Learning, ECCV 2020
>
> [7] The Effect of Diversity in Meta-Learning, AAAI 2023

---

> > ### Author Response · Authors · 2024-11-20
> > **Part IV**
> >
> > ## 7. What is the meaning of Figure 1? What guarantees the orthogonality among the three methods?
> > We apologize for any misunderstanding regarding the orthogonality among the three methods, which was not our intended implication.
> > We have replotted Figure 1 in the revised manuscript to make the main idea more intuitive. The revised figure emphasizes the core concept: starting with a proof that ICL with transformers is expressive enough to encompass typical meta-learners in the learning algorithm space, this paper demonstrates that ICL possesses meta-level deep learning properties. Consequently, deep learning techniques can be adapted to the meta-level to enhance ICL.
> >
> >
> > ## 8. The notion of "optimal" is vague and needs to be better defined within the context of the study
> > This is a reasonable concern, and we greatly appreciate your thoughtful feedback.
> >
> > We have provided a formal definition of optimality in Appendix C of the  revised manuscript.
> >
> > We claim ICL model learns data-dependent optimal learning algorithms (**DDOLA**), which is different and weaker than (true) optimal learning algorithm (**OLA**).
> >
> > **OLA**: Formally, given a finite training set $D_t=${$(x_i,y_i)$} where each sample is i.i.d.: $(x_i,y_i)\sim p(x,y)$, and a testing set $D_v=${$(x_j,y_j)$} following the same distribution,
> > the optimal learning algorithm (OLA) is $g^*=argmin_{g} \mathbb{E}_{(x_j,y_j)\sim p(x,y)}[Prob[g(x_j, D_t )=y_j]]$.
> > In other words, an OLA is a learning algorithm that makes the most “accurate” prediction given a training set and unseen target inputs from the same distribution.
> > It is possible to know the optimal learning algorithm with prior knowledge of $p(x,y)$. For example, OLS is optimal for linear regression with Gaussian noise.
> > In the paper, three types of tasks are generated in designed ways (see Section 4.1). For example, a MatchNet model with certain parameters (where all modules are identity mappings) is the optimal learning algorithm for Task Type I, and similar analyses apply to the other task types and meta-learners.
> >
> >
> > **DDOLA**: However, meta-learners do not have access to the true $p(x,y)$. They only learn to infer $p(x,y)$ from $D_t=${$(x_i,y_i)$} through meta-training, which inevitably introduces variance and bias, making them data-dependent. We define the best that a randomly initialized and meta-trained deep learner can achieve, given a specific meta-training set, as the DDOLA. This can be empirically approximated by meta-training a deep, randomly initialized MatchNet/ProtoNet/CNP on the corresponding meta-training set (for the three task types, respectively).
> >
> > Thus, we claim that the ICL model learns DDOLA through meta-pretraining, as validated by comparisons with the approximations of DDOLA.
> >
> >
> > ## 9. The main text and the appendix feel disconnected, almost as if they belong to separate studies. More details about the task generation process should be included in the main text, not only in the appendix section.
> > Thanks for your suggestions. We will work on improving the structure of the paper to ensure the main text and appendix are more cohesive and interconnected.

---

> > > ### Comment · Reviewer_bz7Z · 2024-11-22
> > > **Response to the authors' comments**
> > >
> > > Thank you for the effort you’ve put into addressing the initial comments. While I acknowledge the improvements made in the revised manuscript and the additional experiments you have conducted, I remain concerned about certain aspects of the study.
> > >
> > >   - The terminology used throughout the paper departs from previous papers in the meta-learning field, and further clarification should be added in the text to improve the clarity of explanation (see, for example, "permutation invariance" and "pre-training").
> > >   - There are still some inconsistencies in the presentation of the math terms. For example, I can see throughout the text that models are sometimes referred to as $g(x)$ and sometimes as $g(x,\theta)$. Furthermore, I believe the theorems from 3.1 to 3.4 are somewhat unnecessary as they are just repetitions with very small differences. You could, for example, summarize them into a single theorem.
> > >   - What's the meaning of the equation in line 799 Appendix C? It seems to me you are minimizing the expected value that the probability of the model's output equals the true label, which doesn't make sense in my opinion.
> > >   - Can you provide more details on how MAML is applied to an in-context learner? It is well known that MAML does not scale with larger architecture [1]. Also, what is the computational complexity of doing so (in terms of time and memory, for example)? Furthermore, what are you learning at the meta-level? It is not clear to me.
> > >  - By comparing the results in Figure 7 with MetaDataset, I see some inconsistencies. With MetaDataset, you clearly show a consistent improvement, regardless of the number of tasks, but with the toy data, it seems there is almost no improvement when the number of tasks increases. Can you clarify this further?
> > >   - I would like to see more details about the task creation mechanism (e.g., which $p(T)$ did you consider) and which rationale/pipeline/previous literature you used.
> > >   - I am still skeptical about the contributions of some claims as they have already been demonstrated and investigated in previous literature (e.g., adaptation works better during evaluation when diverse tasks are involved, curriculum learning helps the model convergence, a meta-learned model adapts faster than standard fine-tuning).
> > >
> > > General comment:\
> > > The initial submission of this manuscript was significantly below the standards expected for this conference. Both the details of the proposed methodologies and experiments, as well as the overall presentation quality, were notably poor.
> > > While the revised manuscript demonstrates some improvement, substantial weaknesses remain. The notation remains unclear, the contributions lack sufficient depth, and the discussion of results does not adequately support the conclusions drawn. The inclusion of new experiments with images, although a potentially interesting addition, feels disconnected from the main scope of the paper and does not integrate well with the rest of the work.
> > > Moreover, I remain unconvinced about the meaningfulness and impact of the M²ICL and curriculum learning experiments, which continue to lack clarity and justification. For these reasons, I have decided to maintain my original score.
> > >
> > >
> > > References:
> > >
> > > [1] Chen, W.Y., Liu, Y.C., Kira, Z., Wang, Y.C., & Huang, J.B. (2019). A Closer Look at Few-shot Classification. In International Conference on Learning Representations.

---

> > > > ### Author Response · Authors · 2024-11-26
> > > > **Thanks for your response. Part I**
> > > >
> > > > Thank you for your additional comments. We sincerely appreciate your dedicated review efforts, which help make this paper more precise and clear for readers specialized in the meta-learning field.
> > > >
> > > >
> > > > ## 1. The terminology used throughout the paper departs from previous papers in the meta-learning field, and further clarification should be added in the text to improve the clarity of explanation (see, for example, "permutation invariance" and "pre-training").
> > > > Thank you for your feedback.
> > > > The primary goal of this work is to understand what ICL models (e.g., LLMs) learn, why they perform well, and how they can be improved. By formulating ICL models as meta-learners and comparing them with typical meta-learners, we aim to provide insights that inspire strategies to enhance ICL.
> > > >
> > > > We note that you are particularly interested in the  M^2 -ICL proposed in Section 5.1. However, we would like to clarify that this is not the main contribution of our paper. In Section 5, we propose improving ICL by transferring deep learning techniques to the meta-level. Specifically, we explore how meta-learning—a well-established deep learning technique—can be applied at the meta-level to support domain-specific LLMs.  M^2 -ICL is presented as a practical example of this idea, building on the understanding developed in Sections 3 and 4 that ICL models learn learning algorithms with data-dependent optimality and generalizability.
> > > >
> > > > To further clarify the terminology and contributions of our work, we recommend reviewing the most related works [2][4][5][7][8], which provide important context and connections to our approach.
> > > >
> > > >
> > > > ## 2. There are still some inconsistencies in the presentation of the math terms. For example, g(x) and g(x;theta). Furthermore, I believe the theorems from 3.1 to 3.4 are somewhat unnecessary as they are just repetitions with very small differences. You could, for example, summarize them into a single theorem.
> > > > We respectfully disagree with the claim regarding inconsistencies in the presentation of mathematical terms and the assertion that theorems 3.1 to 3.4 are unnecessary.
> > > >
> > > > **Inconsistencies of $g(), g(;\theta)$**
> > > >
> > > > In Section 2.0, we explicitly define $g()$ as a learning algorithm, and $g(;\theta)$ as a learning algorithm with learnable parameter $\theta$ (similar with $\omega$ in [10]), which is a meta-learner. And throughout the paper we use these two terms for above meaning consistently.
> > > >
> > > > **Theorem 3.1-3.4**
> > > >
> > > > We present Theorems 3.1 to 3.4 separately because they are independent and represent significant results, despite appearing similar.
> > > >
> > > > ***Independence of Proofs***:
> > > > Each theorem is independently proved and does not rely on the others. As detailed in Appendix D, the proof for ICL performing a specific  learning algorithm involves decomposing  the learning algorithm  into finite conditioned steps, where each step is further proven to be achievable by a transformer layer. The decomposition process is non-trivial and unique to each category of meta-learners.
> > > >
> > > > ***Significance of Individual Results***:
> > > > Demonstrating that ICL can perform specific types of learning algorithms is meaningful and aligns with prior research efforts. For instance, numerous studies focus on whether ICL can perform gradient descent (e.g., [2][3][4][5]). Therefore, presenting Theorem 3.1 independently to cover this result is not redundant.
> > > > - Theorem 3.1: Demonstrates that ICL with transformers can perform any learning algorithm that gradient-based meta-learners can.
> > > > - Theorem 3.2: Demonstrates that ICL with transformers can perform any learning algorithm that pairwise metric-based meta-learners can.
> > > > - Theorem 3.3: Demonstrates that ICL with transformers can perform any learning algorithm that class-prototype metric-based meta-learners can.
> > > > - Theorem 3.4: Demonstrates that ICL with transformers can perform any learning algorithm that amortization-based meta-learners can.
> > > >
> > > > Together, these theorems establish a comprehensive connection between ICL and existing categories of meta-learners, which constitutes the first major contribution of our paper.
> > > >
> > > > ## 3. What's the meaning of the equation in line 799 Appendix C?
> > > > We sincerely apologize for this error and appreciate your careful reading. In line 799, the use of “argmin” was a mistake; it should instead be “argmax.” Correcting this,
> > > > $g^*=\arg\max_{g} \mathbb{E}_{(x_j,y_j)\sim p(x,y)}[Prob[g(x_j, D_t )=y_j]]$ is the natural definition of optimal learning algorithm.

---

> > > > > ### Author Response · Authors · 2024-11-26
> > > > > **Thanks for your response. Part II**
> > > > >
> > > > > ## 4. Can you provide more details on how MAML is applied to an in-context learner? It is well known that MAML does not scale with larger architecture [1]. Also, what is the computational complexity of doing so (in terms of time and memory, for example)? Furthermore, what are you learning at the meta-level? It is not clear to me.
> > > > > **Details of Applying MAML to the ICL Model**
> > > > >
> > > > > We have added an illustrative figure and pseudocode of the algorithm in Appendix I to clarify how MAML is applied to an in-context learner (ICL).
> > > > >
> > > > > One of the most notable advantages of MAML is its model-agnostic nature, which allows us to apply meta-level MAML by using MAML on another meta-learning model, such as ICL. The concepts of typical MAML for supervised learning models and meta-level MAML for ICL models have the following correspondences, effectively bringing the problem setting to one meta-level higher:
> > > > >
> > > > > - **Data Point →  Task**: In the inner loop, during a single forward pass, typical MAML for a supervised learning model takes input features  x  to compute the loss between the prediction and the true label  y , which is used for the inner update. Meta-level MAML on an ICL model takes an entire task as input to compute the autoregressive loss value using Eqn (13), serving the inner update.
> > > > >
> > > > > - **Task →  Domain**: In the outer loop, typical MAML considers a task consisting of a set of data points. Some data points are used for the inner update to obtain the task-adapted model, while the remaining data points evaluate the task-adapted model to compute the loss for the global update. Meta-level MAML treats a domain as consisting of a set of tasks. Some tasks are used for the inner update to obtain the domain-adapted model, and the other tasks evaluate the domain-adapted model for the global update.
> > > > >
> > > > > - **Meta Learning → Meta-Level Meta Learning**: Typical MAML uses a set of tasks for meta-training and a new set of tasks for meta-testing. In contrast, meta-level MAML uses a set of domains for meta-level meta-training and a new set of domains for meta-level meta-testing.
> > > > >
> > > > >
> > > > >
> > > > > **Empirical Computational Complexity on Meta-Dataset**
> > > > >
> > > > > In our experiment using the Meta-Dataset, we meta-level meta-trained  M $^2$-ICL for 2000 epochs with a meta-level meta-learning rate of 0.0005. The maximum GPU memory usage was approximately 69,821 MB. Regarding time consumption, we apologize for not having an exact record, but we estimate that the total training time did not exceed 100 hours.
> > > > >
> > > > > **MAML’s Scaling with Larger Architectures**
> > > > >
> > > > > In [1], the authors present Figure 3, which indicates that in typical meta-learning settings, MAML does not consistently improve performance as the feature extractor becomes deeper. While this observation is reasonable in their context, it is not directly applicable to our problem for the following reason:
> > > > > * **Our explanation of why MAML does not scale with larger architectures observed in [1]:**
> > > > > In the experiments of [1], MAML is the only baseline that adapts all parameters—including the feature extractor—to a few-shot task, whereas other baselines adapt only the classifier while keeping the feature extractor frozen. A deeper feature extractor in MAML may be more prone to overfitting on few-shot tasks, which is a unique risk compared to the other baselines.
> > > > > * **Such reason does not affect our empirical study of meta-level meta-learning.**
> > > > > As mentioned above, MAML's weakness with larger architectures is due its adaptation strategy: fine-tuning all parameters. However, such adaptation strategy is conventional for domain adaptation of ICL models (pre-training fine-tuning). Our study is controlling the variable of adaptation strategy by setting fine-tuning all parameters as default, while modifying the pre-training process to meta-level meta-training.
> > > > >
> > > > > * **MAML does not necessarily suffer scaling with larger architectures.** As mentioned above, MAML's weakness with larger architectures is due fine-tuning a large amount of parameters that easily overfit. However, this could be alleviate by modifying the adaptation strategy. For example, in typical meta-learning, many works only update the parameter of last few layers of the model during inner-update of MAML. In meta-level meta-learning, as discussed in the paper, the ICL model's domain adaptation could be achieved by parameter-efficient approaches such asLORA, prefix-tuning, which can also serve as the inner-update of meta-level MAML.
> > > > > * **Meta-level meta-learning does not necessarily use MAML at meta-level.** The main idea of Section 5.1 in our paper is that one can perform meta-level meta-learning using Eqn (8). We chose the meta-level meta-learner  $G(;\Delta)$  to be MAML due to the reasons discussed around line 458. Importantly, the meta-level meta-learner  $G(;\Delta)$  does not have to be MAML; researchers can design customized  $G(;\Delta)$, such as metric-based or amortization-based meta-learners.

---

> > > > > > ### Author Response · Authors · 2024-11-26
> > > > > > **Thanks for your response. Part III**
> > > > > >
> > > > > > ## 5. By comparing the results in Figure 7 with MetaDataset, I see some inconsistencies. With MetaDataset, you clearly show a consistent improvement, regardless of the number of tasks, but with the toy data, it seems there is almost no improvement when the number of tasks increases. Can you clarify this further?
> > > > > >
> > > > > > We understand that your question could have two interpretations, and we’re not certain which one aligns with your concern:
> > > > > >
> > > > > > **Why don’t the lines in Figure 7(c) show significant improvement compared to Figures 7(a) and 7(b)?**
> > > > > >
> > > > > > Each subfigure in Figure 7 is intended to be considered independently. In Figure 7(a), the blue line represents the best performance, showing improvement over the orange and red lines; the same pattern holds in Figures 7(b) and 7(c). The lines across different subfigures are not directly comparable because each subfigure corresponds to a specific setting with its own pool of training and testing data. Consequently, the absolute values on the y-axis are not comparable across subfigures.
> > > > > >
> > > > > > **Why doesn’t the  M^2 -ICL w/ adpt in Figure 7 show as significant an improvement over ICL w/ adpt as it does in the Meta-Dataset experiments, especially in Figure 7(c)?**
> > > > > >
> > > > > > The difference arises due to varying experimental settings between Figure 7 and the Meta-Dataset. In the Meta-Dataset experiments, we evaluate 8, 16, and 32 tasks-for-adaptation per domain, whereas in Figure 7, we use 64, 256, and 1024 tasks. In typical meta-learning, it is natural that the advantage of MAML over pretraining fine-tuning diminishes as the number of shots increases, due to the reduced risk of overfitting with more data. This tendency similarly appears at the meta-level concerning the number of tasks. We chose 64, 256, and 1024 tasks in Figure 7 to cover a wider range of settings, and 8, 16, and 32 in the Meta-Dataset because real-world domains often have very limited numbers of tasks and computational budget constraints with more complex models.
> > > > > >
> > > > > > ## 6.  I would like to see more details about the task creation mechanism and which rationale/pipeline/previous literature you used.
> > > > > >
> > > > > > We appreciate your attention to detail. The task creation mechanism is provided in Appendix E. While the specific values and distributions are not detailed in the main text, they are available in the supplementary code, as they do not significantly impact our main results.
> > > > > > For example, to create a 2-dimensional binary amortization-based classification task  $\tau$, we first sample  $\mu_\tau \in \mathbb{R}^2$ from $\mathcal{N}(0,I)$. Then, we sample  $N_\tau$ (e.g., 256) i.i.d $x^{(i)}$ from $\mathcal{N}(\mu_\tau,I)$, then assign label $y^{(i)}$ for each $x$ by $\text{sign}(\sum_{d=1}^{2}[x^{(i)}-\mu_\tau]_d)$.
> > > > > >
> > > > > > Similar task creation methods for experiments aimed at understanding ICL have been employed in the literature for regression tasks, such as in [2][3][4][6][7]. We followed the same pipeline and made minor modifications to create tasks with known optimal learning algorithms corresponding to different types of typical meta-learners.

---

> > > > > > > ### Author Response · Authors · 2024-11-26
> > > > > > > **Thanks for your response. Part IV**
> > > > > > >
> > > > > > > ## 7. I am still skeptical about the contributions of some claims as they have already been demonstrated and investigated in previous literature (e.g., adaptation works better during evaluation when diverse tasks are involved, curriculum learning helps the model convergence, a meta-learned model adapts faster than standard fine-tuning).
> > > > > > >
> > > > > > > We would like to emphasize that the main purpose of our work is to understand what In-Context Learning (ICL) models (e.g., Large Language Models or LLMs) learn and why they perform well. We achieve this by formulating them as meta-learners and comparing them with typical meta-learners, which in turn inspires strategies to improve ICL.
> > > > > > >
> > > > > > > We acknowledge that you are particularly interested in the  M $^2$-ICL approach we proposed in Section 5.1. However, we consider this to be a secondary contribution of our paper. In Section 5, our intention is to illustrate how deep learning techniques can be transferred to the meta-level to enhance ICL, with meta-learning serving as an example of such a technique that can help build domain-specific LLMs when applied at the meta-level. This is a practical demonstration of our idea, following our understanding that ICL models learn learning algorithms with data-dependent optimality and generalizability, as discussed in Sections 3 and 4.
> > > > > > > We would like to summarize our specific contributions that distinguish this paper from existing works as follows:
> > > > > > >
> > > > > > > - **Section 3**: We theoretically prove that ICL with transformer is expressive enough to encompass typical meta-learners, comprehensively including gradient-based, metric-based, and amortization-based meta-learners. While existing works have demonstrated this expressiveness for gradient-based meta-learners, our work extends it to cover a broader range.
> > > > > > > - **Section 4**: We show that with sufficient pre-training, ICL models learn data-dependent optimal learning algorithms. These functions implicitly fit the training distribution rather than relying on explicit algorithms, demonstrating the ability to handle novel types of tasks and distribution-sensitive generalization. This challenges existing interpretations of ICL as explicit “algorithm selection,” which have been widely mentioned in previous works.
> > > > > > > - **Section 5**: We propose systematically transferring deep learning techniques to improve ICL by mapping concepts from the supervised learning level to the meta-learning level. This leverages our novel understanding that the ICL model in meta-learning is isomorphic to a deep model in supervised learning. As examples, we implement meta-level meta-learning and meta-level curriculum learning, demonstrating their effectiveness.
> > > > > > >
> > > > > > >
> > > > > > > We believe that reviewing the most related works [2][4][5][7][8] would be very helpful in better understanding our paper, including its terminology and contributions.
> > > > > > >
> > > > > > >
> > > > > > > [1] A Closer Look at Few-shot Classification, ICLR 2019
> > > > > > >
> > > > > > > [2] Transformers Learn In-Context by Gradient Descent, ICML 2023
> > > > > > >
> > > > > > > [3] Transformers learn to implement preconditioned gradient descent for in-context learning, NeurIPS 2023
> > > > > > >
> > > > > > > [4] Transformers as Statisticians: Provable In-Context Learning with In-Context Algorithm Selection, NeurIPS 2023
> > > > > > >
> > > > > > > [5] In-context Learning on Function Classes Unveiled for Transformers, ICML 2024
> > > > > > >
> > > > > > > [6] Pretraining task diversity and the emergence of non-bayesian in-context learning for regression, NeurIPS 2023
> > > > > > >
> > > > > > > [7] What learning algorithm is in-context learning? Investigations with linear models, ICLR 2023
> > > > > > >
> > > > > > > [8] General-purpose in-context learning by meta-learning transformers, 2022
> > > > > > >
> > > > > > > [9] Meta-learning in neural networks: A survey, TPAMI 2020

---

> > > > > > > > ### Author Response · Authors · 2024-12-02
> > > > > > > > **A Sincere Response to Reviewer bz7Z**
> > > > > > > >
> > > > > > > > Dear Reviewer bz7Z,
> > > > > > > >
> > > > > > > > We hope this message finds you well.
> > > > > > > > We appreciate your dedication to the review a lot. Your valuable comments and suggestions did help us improve this paper a lot in some perspectives.
> > > > > > > >
> > > > > > > > However, we have noticed a significant disparity between your evaluation and those of the other reviewers. While you have expressed a strong rejection and described the paper as significantly below the standards expected for this conference, the other reviewers have found thorough theoretical foundation and significant insights about understanding ICL, providing enough contribution to be accepted and evaluated as a good paper.
> > > > > > > >
> > > > > > > > We believe this contradictory situation may stem from a misunderstanding or oversight regarding the main contributions of our paper. To clarify, the primary aim of our work is to understand what In-Context Learning (ICL) models (e.g., Large Language Models or LLMs) learn and why they perform well. We achieve this by formulating ICL models as meta-learners and comparing them with typical meta-learners, thereby inspiring strategies to improve ICL.
> > > > > > > >
> > > > > > > > The specific contributions of this paper can be summarized as follows:
> > > > > > > > - **Section 3**: We theoretically prove that ICL with transformer is expressive enough to encompass typical meta-learners, comprehensively including gradient-based, metric-based, and amortization-based meta-learners. While existing works have demonstrated this expressiveness for gradient-based meta-learners, our work extends it to cover a broader range.
> > > > > > > > - **Section 4**: We show that with sufficient pre-training, ICL models learn data-dependent optimal learning algorithms. These functions implicitly fit the training distribution rather than relying on explicit algorithms, demonstrating the ability to handle novel types of tasks and distribution-sensitive generalization. This challenges the widely accepted interpretation of ICL as explicit “algorithm selection” proposed in previous works.
> > > > > > > > - **Section 5**: We propose systematically transferring deep learning techniques to improve ICL by mapping concepts from supervised learning to meta-learning. This is based on our novel understanding that ICL models in meta-learning are isomorphic to deep models in supervised learning. As examples, we implement meta-level meta-learning and meta-level curriculum learning, demonstrating their effectiveness. We note that much of your feedback centers on meta-level meta-learning, which corresponds to Section 5.1 of the paper. As this section constitutes less than 10% of the paper, its contribution is proportionally minor. Moreover, several of your comments on meta-level meta-learning seem to reflect misunderstandings, which we addressed in detail in our earlier response.
> > > > > > > >
> > > > > > > >
> > > > > > > > We believe your overarching concerns may be influenced by the statement in your review that “there are incorrect assumptions present throughout the paper.” However, we have explained in detail why this concern is not valid in our initial response. We hope you will carefully consider our second response and reevaluate the paper in light of the distinctions between understanding ICL and typical meta-learning approaches, as well as our contributions to the field.
> > > > > > > >
> > > > > > > > We kindly request that you take our responses into consideration alongside the evaluations of the other reviewers. Your acknowledgment of these contributions would mean a great deal to us.
> > > > > > > >
> > > > > > > > Thank you again for your time and dedication to the review process.
> > > > > > > >
> > > > > > > > Best regards,
> > > > > > > > Author

---

> > > > > > > > > ### Comment · Reviewer_bz7Z · 2024-12-02
> > > > > > > > > **Response to authors**
> > > > > > > > >
> > > > > > > > > I thank the authors for their efforts in addressing my concerns. I have updated my score, but I am still unconvinced about the paper's contributions. As noted by other reviewers, although this paper explores various methods within the context of ICL, the contributions appear limited in scope. If the authors intended to demonstrate that ICL fundamentally aligns with meta-learning (learning to learn), a formal definition of the core concepts, accompanied by a stronger theoretical framework, would be necessary to substantiate this claim. Instead, the paper discusses a range of topics, such as meta-level meta-learning and curriculum learning, which come across as supplementary but disconnected contributions.
> > > > > > > > >
> > > > > > > > > There are also other aspects in this paper that I believe should be revisited, or better explain:
> > > > > > > > >   - The meta-learning techniques explored in this paper are relatively old and they might not provide a fair comparison
> > > > > > > > > It is still unclear to me how meta-meta-learning is defined in this paper. From my understanding, the authors directly apply the bi-level optimization of MAML on top of a transformer (claiming that the transformer is acting as a meta-learner). However, this does not constitute meta-meta-learning as understood in the literature. Typically, meta-meta-learning involves learning hyperparameters, structures, or model architecture, which this paper does not address [4]. Additionally, prior research [3] suggests that MAML is unsuitable for large architectures like transformers, raising concerns about the feasibility of the proposed meta-level meta-learning approach.
> > > > > > > > >   - There were some substantial errors in the text and equations in the previous versions of the manuscript, which were fully addressed by the authors. Given the extent of these changes, I am concerned about potential undetected errors in the current version.
> > > > > > > > >   - Interesting results, such as those on Meta-Dataset, are relegated to the appendix without explanation or justification for their inclusion. These results are not even mentioned in the main text and therefore seems completely disconnected from the rest of the paper.
> > > > > > > > >
> > > > > > > > > In conclusions, I recognize the potential of this manuscript, but a significant revision is needed to address the outlined issues before it can be considered for acceptance.
> > > > > > > > >
> > > > > > > > > References:\
> > > > > > > > > [1] Sang Michael Xie and Sewon Min. How does in-context learning work? a framework for understanding the differences from traditional supervised learning. Online, 2022. 21.\
> > > > > > > > > [2] Brown, Tom B. "Language models are few-shot learners." arXiv preprint arXiv:2005.14165 (2020).\
> > > > > > > > > [3] Chen, Wei-Yu, et al. "A Closer Look at Few-shot Classification." International Conference on Learning Representations. 2019.\
> > > > > > > > > [4] Vettoruzzo, Anna, et al. "Advances and challenges in meta-learning: A technical review." IEEE Transactions on Pattern Analysis and Machine Intelligence (2024).

---

> > > > > > > > > > ### Author Response · Authors · 2024-12-03
> > > > > > > > > > **Thanks for Your Response**
> > > > > > > > > >
> > > > > > > > > > Thank you for your response. We are sorry to find that you did not fully enjoy or grasp the main ideas of our paper, but we sincerely appreciate the time and effort you dedicated to reviewing it.

---

### Official Review · Reviewer_prfg · 2024-10-26

**Soundness:** 4
**Presentation:** 4
**Contribution:** 3
**Rating:** 8
**Confidence:** 4

**Summary:**

Summary:
This paper studies in-context learning (ICL) models from the perspective of meta-learning with solid theoretical analysis and experiments.

Contributions:
1.It theoretically proved that the hypothesis space of ICL encompass the hypothesis spaces of typical meta-learners.
2. it experimentally verified that ICL with transformer learns the optimal algorithms by designing different types of meta-learning tasks, and several other aspects.
3.It proposed new techniques including meta-level meta-learning and meta-level curriculum learning to improve ICL’s performance and convergence, respectively.

**Strengths:**

1.The paper is well-motivated, well-written and easy to follow.
2.The topic discussed in this paper is interesting and worth studying.
3.The proposed assumptions are proved and confirmed with solid theoretical analysis and experiments.
4.The contribution of this paper is solid, please refer to the summary of contributions.

**Weaknesses:**

1.The paper only focuses on ICL with transformer, ICL with other deep architectures are also worth exploring
2.Typo: line 482 tunie

**Questions:**

1.Please elaborate why M^2-ICL without adaptation sometimes works worse than ICL
2.For meta-level curriculum learning, how would you define the complexity of other tasks, such as classification tasks? Could you add some more experiments with regard to few-shot classification tasks?
3.For task generation, what is the definition of p_f, and how to obtain p_f for different tasks?

---

> ### Author Response · Authors · 2024-11-20
> **Part I**
>
> Thanks for the valuable comments and suggestions! We are delighted that you like this paper.
>
> ## W1. Explore ICL with deep architectures other than transformer
> Indeed, as discussed in Section 6, exploring ICL with other deep architectures, such as SSMs and RNNs [1][2], is a worthwhile direction and has been the focus of some works.
>
> In this paper, we focus on transformers as they are the most common model for ICL and are closely aligned with our primary goal: explaining the success of large language models (LLMs).
>
> The results presented in this paper might be generalizable to ICL with other architectures. While the theoretical proof of expressiveness in Section 3 is specific to transformers, similar approaches for proving the expressiveness of label-aware set functions could potentially be extended to other architectures. Additionally, the empirical results in Section 4 and the proposed methods in Section 5 are likely applicable to ICL with other architectures, as they rely on deep-parameterized meta-learners rather than being strictly transformer-based.
>
> [1] Mamba: Linear-Time Sequence Modeling with Selective State Spaces
>
> [2] IS ATTENTION REQUIRED FOR ICL? EXPLORING THE RELATIONSHIP BETWEEN MODEL ARCHITECTURE AND IN-CONTEXT LEARNING ABILITY, ICLR 2024
>
> ## W2. Typo
> Thanks a lot for your careful reading. We apologize for the oversight and have corrected the typos in the revised manuscript.
>
> ## Q1. Please elaborate why M^2-ICL without adaptation sometimes works worse than ICL
> Please note that we cannot conclusively determine whether  M^2 -ICL w/o adpt works better or worse than ICL w/o adpt. ICL is trained for the global objective (Eq. 13), while  M^2 -ICL is trained for the post-adaptation objective (Eq. 8). Their comparison is analogous to comparing a model trained using standard supervised learning with one meta-trained using MAML on tasks drawn from the same training set. When tested on novel classes (unseen tasks) without adaptation, neither approach is expected to perform well.
>
> When a few labeled data points are available for adaptation, reasonable strategies include:
>
> - **ICL w/ adpt**: May perform well when sufficient data is available but is prone to overfitting when data is limited.
> - **ICL w/o adpt**: Avoids overfitting but may struggle when more data is available.
> - **M^2 -ICL w/ adpt**: Designed to address overfitting by modifying the pretraining process, making it better suited for scenarios with very few labeled data points.
>
> The meta-level meta-learning approach is specifically designed for  M^2 -ICL w/ adpt.  M^2 -ICL w/o adpt is only an intermediate product, and its main role is to serve as a baseline for comparison with  M^2 -ICL w/ adpt, to verify that adaptation benefits  M^2 -ICL even with very limited data.
>
> We have made minor modifications to the explanation of these results in the revised manuscript to improve clarity and understanding (line 475-477, 484-485, in blue color).
>
> ## Q2. How to define the complexity of other tasks, such as classification tasks? Add some more experiments with regard to few-shot classification tasks (common)
> The concern about real-world applications of the proposed strategies is reasonable.
>
> **Meta-level curriculum-learning:**
>
> For meta-level curriculum learning, there are existing works that demonstrate practical approaches with typical meta-learners on real-world problems. There are various strategies to determine which tasks should be learned in a given episode during meta-training. A common approach is knowledge-based curriculum learning, where task difficulty is defined using expert knowledge specific to the problem. For instance, [2] defines task difficulty based on interactions between semantically similar relations observed in a task sequence for continual relation extraction problems.
> Another approach is model-based curriculum learning, where task difficulty is determined based on the evaluation performance of the current model, as explored in [3][4]. More generally, to schedule task-sampling strategies in meta-learning, additional factors such as task diversity within a batch or other task-specific information can be considered, as proposed in [5][6][7].
> These strategies can be extended to the pretraining of ICL models for real-world problems by replacing the meta-learner with the ICL model. We believe adopting such strategies could help construct an effective curriculum for complex, real-world datasets.

---

> > ### Comment · Reviewer_prfg · 2024-12-01
> > **Thank you for the response**
> >
> > The comments address my concerns, I maintain my original score.

---

> ### Author Response · Authors · 2024-11-20
> **Part II**
>
> **Meta-level meta-learning:**
>
> We have evaluated the effectiveness of meta-level meta-learning on a real-world few-shot image classification problem using the Meta-Dataset [1]. This dataset is particularly suitable as it contains multiple sub-datasets, each of which can be used to sample many few-shot classification tasks, making it naturally divisible into multiple domains for meta-level meta-training (Eqn (8)).
>
> Following standard settings, we used the training sets of ILSVRC, Omniglot, Aircraft, Birds, Textures, Quick Draw, and Fungi during training. For testing, we used unseen datasets such as Traffic Signs, MSCOCO, and additional datasets like MNIST, CIFAR10, and CIFAR100. Each dataset is treated as a domain for meta-level meta-learning.
> We considered 5-way 5-shot tasks at the meta-level and 8/16/32 tasks-for-adaptation per domain at the meta-meta-level. The sampling of classes and images to form tasks, as well as the sampling of tasks-for-adaptation within a domain, was random.
>
>
> We consider the following baselines:
> *  **ICL w/o adpt**: The standard meta-learning setting, where meta-training is performed on all tasks without distinguishing between datasets. During meta-testing, no domain adaptation is performed, meaning that the 8/16/32 tasks are not utilized.
> *  **ICL w/ adpt**: The meta-training process is identical to that of ICL w/o adpt. While during meta-testing, the model adapts using 8/16/32 domain-specific tasks by fine-tuning all parameters (step = 5, learning rate = 0.0001, batch size = 8/16/32).
> *  **M $^2$-ICL**: Meta-level meta-training of an ICL model using the method introduced in Section 5.1. The domain adaptation process (i.e., the inner loop of meta-level MAML) is configured with step = 5, learning rate = 0.0001, and 16 tasks-for-adaptation per domain. Meta-meta-training and evaluation were conducted for 8/16/32 tasks-for-adaptation, respectively. During testing, the same adaptation process was applied.
>
> Our implementation utilized an 8-layer transformer (without positional encoding) as the ICL model, with input features extracted by a ResNet-18 (512 dimensions). Although this model is relatively simple, it is sufficient to verify the effectiveness of meta-level meta-learning in improving ICL for real-world applications. The pipeline is generalizable, and the architecture can be replaced with more advanced models or adapted for other applications.
>
> The results, presented in the table below and inlcuded in Appendix H of the revised manuscript, demonstrate that M $^2$-ICL significantly outperforms both ICL w/o adpt and ICL w/ adpt across all task settings. Specifically, adapting the ICL model with only 8 tasks often results in overfitting, while using 16 tasks shows limited improvement, and 32 tasks leads to marginal gains. In contrast, adapting M $^2$-ICL with just 8 tasks surpasses the average performance of other methods, and increasing the number of tasks for adaptation brings further improvements. Notably, ICL w/o adpt, ICL w/ adpt(32 tasks), and M $^2$-ICL (8 tasks) show comparable performance. These results highlight the effectiveness of the proposed meta-level meta-learning approach in enhancing few-task domain adaptation.
>
> Method | Traffic Signs| MSCOCO | MNIST | CIFAR10| CIFAR100 | Average
> ------------- | -------------|--|--|--|--|--|
> ICL w/o adpt  | 45.4 |35.5|88.1 |65.2|55.9 |58.02
> ICL w/ adpt (8 tasks) | 41.9 |35.1|76.4 |64.9|55.5|53.76
> ICL w/ adpt (16 tasks) | 43.3|36.2|78.5|66.0|56.3|56.06
> ICL w/ adpt (32 tasks) | 46.1|36.5|83.2|66.8|58.3|58.18
> M $^2$-ICL (8 tasks)| 45.9|39.4|86.6|67.4|57.2|59.30
> M $^2$-ICL (16 tasks)| 47.5|40.6|88.9|68.0|57.9|60.58
> M $^2$-ICL (32 tasks)| **52.6**|**44.1**|**91.0**|**69.4**|**59.2**|**63.26**
>
> [1] Meta-Dataset: A dataset of datasets for learning to learn from few examples, ICLR 2020
>
> [2] Curriculum-meta learning for order-robust continual relation extraction, AAAI 2021
>
> [3] Curriculum meta-learning for next POI recommendation, KDD 21
>
> [4] Progressive Meta-Learning With Curriculum, 2022
>
> [5] Meta-Curriculum Learning for Domain Adaptation in Neural Machine Translation, AAAI 2021
>
> [6] Adaptive Task Sampling for Meta-Learning, ECCV 2020
>
> [7] The Effect of Diversity in Meta-Learning, AAAI 2023
>
>
> ##  Q3. p_f in task generation
> The $p_f$ in line 384 should be $p(x)$.
> We apologize for the notation error and greatly appreciate your careful reading. This has been corrected in the revised manuscript. As introduced in Appendix B.1 and Appendix E, we sample tasks (functions) from $p(\tau)$ and inputs from $p(x)$ to build a task $D_\tau$.

---

### Official Review · Reviewer_APrh · 2024-11-03

**Soundness:** 3
**Presentation:** 3
**Contribution:** 4
**Rating:** 6
**Confidence:** 4

**Summary:**

This paper investigates ICL model from a learning-to-learn perspective, examining its expressiveness, learnability and generalizability as a meta-learner. It theoretically demonstrates that ICL, when integrated with a transformer, is expressive to perform various existing categories of meta-learning algorithms. The paper also reveals that while the ICL model can learn data-dependent optimal algorithms, but it is not "algorithm selection". Finally, it proposes training ICL models using a task-level meta-learning framework and with curriculum.

**Strengths:**

1. The exploration of the ICL model from a learning-to-learn perspective is thorough and yields significant insights, particularly in enhancing our understanding of ICL's role in large transformers. Notably, the clarification that "ICL is not algorithm selection" challenges previous interpretations and adds depth to the academic discussion on this topic.
2. The clarity and logical flow in Sections 3 and 4 are particularly commendable, making complex findings accessible and substantiating the solidity of the research.

**Weaknesses:**

1. The introduction section could be improved for better accessibility. Key concepts such as "expressiveness," "learnability," and "generalizability" should be clearly defined early. For instance, integrating these definitions in the introduction section would enhance comprehension and engagement from the outset. Additionally, Figure 1 needs a clearer explanation to effectively convey its intended message.
2. Another concern is the validation of the proposed methods, which currently relies solely on simplistic synthetic data. This limitation raises questions about the practical applicability of the methods in real-world scenarios.

**Questions:**

Could the authors elaborate on whether the proposed methods can be adapted for use with real-world datasets? If so, what modifications or considerations would be necessary to accommodate the complexities of real-world data?

---

> ### Author Response · Authors · 2024-11-20
> **Part I**
>
> Thanks for the valuable comments and suggestions! We are glad that you find some parts of our paper commendable.
> ## W1. The introduction section could be improved for better accessibility
> Thanks for the kind suggestion. We have replotted Fig 1 to make the main idea more intuitive, and would further emphasize in later version of the manuscript. The revised parts in pdf are highlighted in blue.
>
> ## W2&Q. Could the authors elaborate on whether the proposed methods can be adapted for use with real-world datasets? If so, what modifacations or considerations would be necessary to accommodate the complexities of real-world data?
> Yes, our proposed methods can be applied for real-world datasets.
>
> **Meta-level meta-learning:**
>
> We have evaluated the effectiveness of meta-level meta-learning on a real-world few-shot image classification problem using the Meta-Dataset [1]. This dataset is particularly suitable as it contains multiple sub-datasets, each of which can be used to sample many few-shot classification tasks, making it naturally divisible into multiple domains for meta-level meta-training (Eqn (8)).
>
> Following standard settings, we used the training sets of ILSVRC, Omniglot, Aircraft, Birds, Textures, Quick Draw, and Fungi during training. For testing, we used unseen datasets such as Traffic Signs, MSCOCO, and additional datasets like MNIST, CIFAR10, and CIFAR100. Each dataset is treated as a domain for meta-level meta-learning.
> We considered 5-way 5-shot tasks at the meta-level and 8/16/32 tasks-for-adaptation per domain at the meta-meta-level. The sampling of classes and images to form tasks, as well as the sampling of tasks-for-adaptation within a domain, was random.
>
>
> We consider the following baselines:
> *  **ICL w/o adpt**: The standard meta-learning setting, where meta-training is performed on all tasks without distinguishing between datasets. During meta-testing, no domain adaptation is performed, meaning that the 8/16/32 tasks are not utilized.
> *  **ICL w/ adpt**: The meta-training process is identical to that of ICL w/o adpt. While during meta-testing, the model adapts using 8/16/32 domain-specific tasks by fine-tuning all parameters (step = 5, learning rate = 0.0001, batch size = 8/16/32).
> *  **M $^2$-ICL**: Meta-level meta-training of an ICL model using the method introduced in Section 5.1. The domain adaptation process (i.e., the inner loop of meta-level MAML) is configured with step = 5, learning rate = 0.0001, and 16 tasks-for-adaptation per domain. Meta-meta-training and evaluation were conducted for 8/16/32 tasks-for-adaptation, respectively. During testing, the same adaptation process was applied.
>
> Our implementation utilized an 8-layer transformer (without positional encoding) as the ICL model, with input features extracted by a ResNet-18 (512 dimensions). Although this model is relatively simple, it is sufficient to verify the effectiveness of meta-level meta-learning in improving ICL for real-world applications. The pipeline is generalizable, and the architecture can be replaced with more advanced models or adapted for other applications.
>
> The results, presented in the table below and inlcuded in Appendix H of the revised manuscript, demonstrate that M $^2$-ICL significantly outperforms both ICL w/o adpt and ICL w/ adpt across all task settings. Specifically, adapting the ICL model with only 8 tasks often results in overfitting, while using 16 tasks shows limited improvement, and 32 tasks leads to marginal gains. In contrast, adapting M $^2$-ICL with just 8 tasks surpasses the average performance of other methods, and increasing the number of tasks for adaptation brings further improvements. Notably, ICL w/o adpt, ICL w/ adpt(32 tasks), and M $^2$-ICL (8 tasks) show comparable performance. These results highlight the effectiveness of the proposed meta-level meta-learning approach in enhancing few-task domain adaptation.
>
> Method | Traffic Signs| MSCOCO | MNIST | CIFAR10| CIFAR100 | Average
> ------------- | -------------|--|--|--|--|--|
> ICL w/o adpt  | 45.4 |35.5|88.1 |65.2|55.9 |58.02
> ICL w/ adpt (8 tasks) | 41.9 |35.1|76.4 |64.9|55.5|53.76
> ICL w/ adpt (16 tasks) | 43.3|36.2|78.5|66.0|56.3|56.06
> ICL w/ adpt (32 tasks) | 46.1|36.5|83.2|66.8|58.3|58.18
> M $^2$-ICL (8 tasks)| 45.9|39.4|86.6|67.4|57.2|59.30
> M $^2$-ICL (16 tasks)| 47.5|40.6|88.9|68.0|57.9|60.58
> M $^2$-ICL (32 tasks)| **52.6**|**44.1**|**91.0**|**69.4**|**59.2**|**63.26**

---

> > ### Author Response · Authors · 2024-11-20
> > **Part II**
> >
> > **Meta-level curriculum learning:**
> >
> > For meta-level curriculum learning, there are existing works that demonstrate practical approaches with typical meta-learners on real-world problems. There are various strategies to determine which tasks should be learned in a given episode during meta-training. A common approach is knowledge-based curriculum learning, where task difficulty is defined using expert knowledge specific to the problem. For instance, [2] defines task difficulty based on interactions between semantically similar relations observed in a task sequence for continual relation extraction problems.
> > Another approach is model-based curriculum learning, where task difficulty is determined based on the evaluation performance of the current model, as explored in [3][4]. More generally, to schedule task-sampling strategies in meta-learning, additional factors such as task diversity within a batch or other task-specific information can be considered, as proposed in [5][6][7].
> > These strategies can be extended to the pretraining of ICL models for real-world problems by replacing the meta-learner with the ICL model. We believe adopting such strategies could help construct an effective curriculum for complex, real-world datasets.
> >
> >
> > [1] Meta-Dataset: A dataset of datasets for learning to learn from few examples, ICLR 2020
> >
> > [2] Curriculum-meta learning for order-robust continual relation extraction, AAAI 2021
> >
> > [3] Curriculum meta-learning for next POI recommendation, KDD 21
> >
> > [4] Progressive Meta-Learning With Curriculum, 2022
> >
> > [5] Meta-Curriculum Learning for Domain Adaptation in Neural Machine Translation, AAAI 2021
> >
> > [6] Adaptive Task Sampling for Meta-Learning, ECCV 2020
> >
> > [7] The Effect of Diversity in Meta-Learning, AAAI 2023

---

> ### Comment · Reviewer_APrh · 2024-11-25
> **Thanks for the response from the authors**
>
> Thank you to the authors for the response. I maintain my score to support the acceptance of this paper for its valuable empirical studies on ICL and the insights these studies provide into the learning processes involved.

---

> > ### Author Response · Authors · 2024-11-26
> > **Thanks, a new pdf uploaded!**
> >
> > Thank you for your positive feedback and for recognizing the value of our additional empirical studies. We are delighted that you found the insights into the learning processes involved to be valuable. We have updated our manuscript for better accessibility. Please let us know if there are any further concern or related question. Your support is greatly important and appreciated.

---

### Official Review · Reviewer_CiDY · 2024-11-09

**Soundness:** 3
**Presentation:** 3
**Contribution:** 3
**Rating:** 6
**Confidence:** 2

**Summary:**

The paper begins by framing the rise of large language models (LLMs) and their use of in-context learning (ICL) to perform diverse tasks without fine-tuning. This sets up ICL as a unique form of "learning to learn," contrasting with traditional meta-learning models. A theoretical foundation is provided, showing that ICL, particularly with transformers, can express and replicate the behaviors of all main meta-learning approaches (gradient-based, metric-based, and amortization-based). The paper tests ICL’s ability to learn optimal algorithms in three task types, showing that it can indeed achieve data-dependent optimal performance on simple, homogeneous tasks. However, when tasked with mixed types, ICL models exhibit implicit (distribution-sensitive) learning rather than general, explicit algorithm selection. This distinction introduces the core limitation: limited generalizability. To address generalizability and efficiency issues, the authors suggest transferring established deep-learning techniques into the meta-learning realm. They propose meta-level meta-learning for domain adaptability (training ICL models to quickly adapt to new domains with limited data) and meta-level curriculum learning to accelerate pre-training.

**Strengths:**

- The paper provides a theoretical foundation for understanding ICL models as learning algorithms, proving their expressiveness and comparing them to traditional meta-learning algorithms.
- By comparing ICL with various meta-learning techniques, it offers insights into how and why ICL might outperform traditional meta-learning methods due to its flexibility in the hypothesis space.
- The paper highlights the critical issue of generalizability in ICL models and offers potential solutions to improve it.

**Weaknesses:**

- The extensive theoretical background may limit accessibility for practitioners interested in practical applications, as certain concepts (e.g., implicit vs. explicit optimality) are dense and could benefit from simpler, more intuitive explanations.
- The proposed solutions, meta-level meta learning and meta-level curriculum learning, are promising but lack empirical evaluation on complex, real-world tasks where curriculum ordering could be less clear-cut. Researchers are suing ICL since it can save us from training tons of parameters, although propose method can improve ICL’s performance on specific domain with very limited data for adaptation and transferring deep-learning techniques to the meta-level is a good point, the proposed method lack its practicability.
- There is a vague treatment of data dependency in "optimal" algorithms. The paper mentions data-dependent optimality without fully clarifying how this dependency impacts performance across tasks. The authors do not delve into specific criteria or metrics for measuring optimality across distributions, making it challenging to objectively evaluate their claims.

**Questions:**

- How would you define or quantify the "optimality" of the ICL model’s learning algorithm in real-world settings with diverse and complex task distributions?
- ICL demonstrates limited generalizability on tasks from seen types, and is sensitive to the data distribution, why this behavior (distribution-sensitiev) is called as "implicit"? How the term "implicit" is defined in this paper?
- Does ICL prioritize simpler or more complex tasks, and what strategies might be implemented to balance performance across tasks of differing complexity levels? Given the challenges of defining a task "difficulty" hierarchy in real-world scenarios, how would you propose constructing an effective curriculum for complex, real-world datasets?
- Does increasing the size or diversity of the training dataset systematically improve generalization across unseen distributions?

---

> ### Author Response · Authors · 2024-11-20
> **Part I**
>
> Thanks for the valuable comments and suggestions!
> ## W1. Simpler and more intuitive explanations for theoretical background
> Thank you for the insightful suggestion. To address this, we have replotted Figure 1 to make the main idea more intuitive. The revised figure emphasizes the core concept: starting with a proof that ICL with transformers is expressive enough to encompass typical meta-learners in the learning algorithm space, this paper demonstrates that ICL possesses meta-level deep learning properties. Consequently, deep learning techniques can be adapted to the meta-level to enhance ICL. The revised sentences in pdf are highlighted in blue.
>
> ## W2. Lack empirical evaluation on real-world task of proposed method.
> Thanks for the suggestion.
> We have evaluated the effectiveness of meta-level meta-learning on a real-world few-shot image classification problem using the Meta-Dataset [1]. This dataset is particularly suitable as it contains multiple sub-datasets, each of which can be used to sample many few-shot classification tasks, making it naturally divisible into multiple domains for meta-level meta-training (Eqn (8)).
>
> Following standard settings, we used the training sets of ILSVRC, Omniglot, Aircraft, Birds, Textures, Quick Draw, and Fungi during training. For testing, we used unseen datasets such as Traffic Signs, MSCOCO, and additional datasets like MNIST, CIFAR10, and CIFAR100. Each dataset is treated as a domain for meta-level meta-learning.
> We considered 5-way 5-shot tasks at the meta-level and 8/16/32 tasks-for-adaptation per domain at the meta-meta-level. The sampling of classes and images to form tasks, as well as the sampling of tasks-for-adaptation within a domain, was random.
>
>
> We consider the following baselines:
> *  **ICL w/o adpt**: The standard meta-learning setting, where meta-training is performed on all tasks without distinguishing between datasets. During meta-testing, no domain adaptation is performed, meaning that the 8/16/32 tasks are not utilized.
> *  **ICL w/ adpt**: The meta-training process is identical to that of ICL w/o adpt. While during meta-testing, the model adapts using 8/16/32 domain-specific tasks by fine-tuning all parameters (step = 5, learning rate = 0.0001, batch size = 8/16/32).
> *  **M $^2$-ICL**: Meta-level meta-training of an ICL model using the method introduced in Section 5.1. The domain adaptation process (i.e., the inner loop of meta-level MAML) is configured with step = 5, learning rate = 0.0001, and 16 tasks-for-adaptation per domain. Meta-meta-training and evaluation were conducted for 8/16/32 tasks-for-adaptation, respectively. During testing, the same adaptation process was applied.
>
> Our implementation utilized an 8-layer transformer (without positional encoding) as the ICL model, with input features extracted by a ResNet-18 (512 dimensions). Although this model is relatively simple, it is sufficient to verify the effectiveness of meta-level meta-learning in improving ICL for real-world applications. The pipeline is generalizable, and the architecture can be replaced with more advanced models or adapted for other applications.
>
> The results, presented in the table below and inlcuded in Appendix H of the revised manuscript, demonstrate that M $^2$-ICL significantly outperforms both ICL w/o adpt and ICL w/ adpt across all task settings. Specifically, adapting the ICL model with only 8 tasks often results in overfitting, while using 16 tasks shows limited improvement, and 32 tasks leads to marginal gains. In contrast, adapting M $^2$-ICL with just 8 tasks surpasses the average performance of other methods, and increasing the number of tasks for adaptation brings further improvements. Notably, ICL w/o adpt, ICL w/ adpt(32 tasks), and M $^2$-ICL (8 tasks) show comparable performance. These results highlight the effectiveness of the proposed meta-level meta-learning approach in enhancing few-task domain adaptation.
>
> Method | Traffic Signs| MSCOCO | MNIST | CIFAR10| CIFAR100 | Average
> ------------- | -------------|--|--|--|--|--|
> ICL w/o adpt  | 45.4 |35.5|88.1 |65.2|55.9 |58.02
> ICL w/ adpt (8 tasks) | 41.9 |35.1|76.4 |64.9|55.5|53.76
> ICL w/ adpt (16 tasks) | 43.3|36.2|78.5|66.0|56.3|56.06
> ICL w/ adpt (32 tasks) | 46.1|36.5|83.2|66.8|58.3|58.18
> M $^2$-ICL (8 tasks)| 45.9|39.4|86.6|67.4|57.2|59.30
> M $^2$-ICL (16 tasks)| 47.5|40.6|88.9|68.0|57.9|60.58
> M $^2$-ICL (32 tasks)| **52.6**|**44.1**|**91.0**|**69.4**|**59.2**|**63.26**
>
> [1] Meta-Dataset: A dataset of datasets for learning to learn from few examples, ICLR 2020

---

> ### Author Response · Authors · 2024-11-20
> **Part II**
>
> ## W3&Q1. Define or quantify "optimality"
> We have provided a formal definition of optimality in Appendix C of the  revised manuscript.
>
> We claim ICL model learns data-dependent optimal learning algorithms (**DDOLA**), which is different and weaker than (true) optimal learning algorithm (**OLA**).
>
> **OLA**: Formally, given a finite training set $D_t=${$(x_i,y_i)$} where each sample is i.i.d.: $(x_i,y_i)\sim p(x,y)$, and a testing set $D_v=${$(x_j,y_j)$} following the same distribution,
> the optimal learning algorithm (OLA) is $g^*=argmax_{g} \mathbb{E}_{(x_j,y_j)\sim p(x,y)}[Prob[g(x_j, D_t )=y_j]]$.
> In other words, an OLA is a learning algorithm that makes the most “accurate” prediction given a training set and unseen target inputs from the same distribution.
> It is possible to know the optimal learning algorithm with prior knowledge of $p(x,y)$. For example, OLS is optimal for linear regression with Gaussian noise.
> In the paper, three types of tasks are generated in designed ways (see Section 4.1). For example, a MatchNet model with certain parameters (where all modules are identity mappings) is the optimal learning algorithm for Task Type I, and similar analyses apply to the other task types and meta-learners.
>
>
> **DDOLA**: However, meta-learners do not have access to the true $p(x,y)$. They only learn to infer $p(x,y)$ from $D_t=${$(x_i,y_i)$} through meta-training, which inevitably introduces variance and bias, making them data-dependent. We define the best that a randomly initialized and meta-trained deep learner can achieve, given a specific meta-training set, as the DDOLA. This can be empirically approximated by meta-training a deep, randomly initialized MatchNet/ProtoNet/CNP on the corresponding meta-training set (for the three task types, respectively).
>
> Thus, we claim that the ICL model learns DDOLA through meta-pretraining, as validated by comparisons with the approximations of DDOLA.
>
> ## Q2. The meaning of "implicit" in this paper
> The terms “implicit” and “explicit” describe the generalizability of a learning algorithm, as introduced in Section 4.
> An implicit learning algorithm is sensitive to the data distribution, while an explicit learning algorithm is insensitive. Most human-designed algorithms, such as OLS or kNN, can generalize to learn any function within a predefined family regardless of the example distribution. These algorithms can typically be expressed using equations and operations with few parameters left to be determined, which is why they are called “explicit.”
>
> In contrast, implicit algorithms have minimal prior assumptions about the function family patterns and rely heavily on parameters that are determined in a data-distribution-dependent manner. Such algorithms are often difficult for humans to explain, hence the term “implicit.”

---

> > ### Author Response · Authors · 2024-11-20
> > **Part III**
> >
> > ## Q3. Does ICL prioritize simpler or more complex tasks, and what strategies might be implemented to balance performance across tasks of differing complexity levels? How to construct an effective curriculum for complex, real-world datasets?
> > This is a good question.
> > We think ICL models tend to prioritize tasks that contribute more significantly to the reduction of the global loss, like other deep models optimized by gradient descent.
> > From our experiments, we have not observed evidence suggesting that ICL behaves differently from conventional deep models in this regard.
> >
> > For meta-level curriculum learning, there are existing works that demonstrate practical approaches with typical meta-learners on real-world problems. There are various strategies to determine which tasks should be learned in a given episode during meta-training. A common approach is knowledge-based curriculum learning, where task difficulty is defined using expert knowledge specific to the problem. For instance, [1] defines task difficulty based on interactions between semantically similar relations observed in a task sequence for continual relation extraction problems.
> > Another approach is model-based curriculum learning, where task difficulty is determined based on the evaluation performance of the current model, as explored in [2][3]. More generally, to schedule task-sampling strategies in meta-learning, additional factors such as task diversity within a batch or other task-specific information can be considered, as proposed in [4][5][6].
> >
> > These strategies can be extended to the pretraining of ICL models for real-world problems by replacing the meta-learner with the ICL model. We believe adopting such strategies could help construct an effective curriculum for complex, real-world datasets.
> >
> > [1] Curriculum-meta learning for order-robust continual relation extraction, AAAI 2021
> >
> > [2] Curriculum meta-learning for next POI recommendation, KDD 21
> >
> > [3] Progressive Meta-Learning With Curriculum, 2022
> >
> > [4] Meta-Curriculum Learning for Domain Adaptation in Neural Machine Translation, AAAI 2021
> >
> > [5] Adaptive Task Sampling for Meta-Learning, ECCV 2020
> >
> > [6] The Effect of Diversity in Meta-Learning, AAAI 2023
> >
> > ## Q4. Does increasing the size or diversity of the training dataset systematically improve generalization across unseen distributions?
> > Yes, this is a good point. Increasing the number and diversity of training tasks does indeed help improve generalization. Quantitative analyses supporting this claim can be found in works such as [1][2].
> > As we had discussed at line 534-537, developing a neural scaling law for ICL to systematically describe the relationships among pretraining size and diversity, generalizability, and model size is an exciting direction for future research. However, this is beyond the scope of the current paper.
> >
> > [1] Pretraining task diversity and the emergence of non-bayesian in-context learning for regression, NeurIPS 2023
> >
> > [2] How Many Pretraining Tasks Are Needed for In-Context Learning of Linear Regression?, ICLR 2024

---

> > > ### Comment · Reviewer_CiDY · 2024-11-25
> > > **Thanks for author's response**
> > >
> > > Thank you for addressing my concerns. This paper ambitiously integrates a range of complex research topics, including meta-learning, adaptation, and curriculum learning, within the broader framework of in-context learning. While the effort to combine these varied themes is commendable, it remains unclear whether each topic introduces novel insights or extends beyond existing research, a concern that other reviewers have also raised. Given the wide array of concepts covered, I recommend that the authors provide more detailed explanations and clearly delineate their novel contributions in relation to prior works.

---

> > > > ### Author Response · Authors · 2024-11-26
> > > > **Thanks and Clarify Contributions**
> > > >
> > > > Thank you for your feedback. We would like to summarize our specific contributions that distinguish this paper from existing works as follows:
> > > >
> > > > - **Section 3**: We theoretically prove that In-Context Learning (ICL) with transformers is expressive enough to encompass typical meta-learners, comprehensively including gradient-based, metric-based, and amortization-based meta-learners. While existing works have demonstrated this expressiveness for gradient-based meta-learners, our work extends it to cover a broader range.
> > > > - **Section 4**: We show that with sufficient pre-training, ICL models learn data-dependent optimal learning algorithms. These functions implicitly fit the training distribution rather than relying on explicit algorithms, demonstrating the ability to handle novel types of tasks and distribution-sensitive generalization. This challenges existing interpretations of ICL as explicit “algorithm selection,” which have been widely mentioned in previous works.
> > > > - **Section 5**: We propose systematically transferring deep learning techniques to improve ICL by mapping concepts from the supervised learning level to the meta-learning level. This leverages our novel understanding that the ICL model in meta-learning is isomorphic to a deep model in supervised learning. As examples, we implement meta-level meta-learning and meta-level curriculum learning, demonstrating their effectiveness.

---

### Meta-Review · Area_Chair_Z25A · 2024-12-16

**Metareview:**

In this paper, the authors investigated ICL from a learning-to-learn perspective, theoretically analyzed its expressiveness with transformers, and presented potential solutions to improve the performance of existing ICML models based on the theoretical findings.

This paper provides new insights into ICL and offers solutions based on those insights. The rebuttal addresses most of the reviewers' concerns. Overall, this paper has sufficient merits and deserves to be published.

**Additional Comments On Reviewer Discussion:**

The negative reviewer was most focused on whether the meta-learning techniques explored in the paper are SOTA during rebuttal. However, analyzing ICL from the meta-learning or learning-to-learn perspective is an interesting idea and it does provide some new insights into ICL. In addition, the reviewer is not working on ICL. Therefore, I think her evaluation is a bit biased.

---

### Decision · Program_Chairs · 2025-01-22

Accept (Poster)